# Effects of spatial resolution on WRF v3.8.1 simulated meteorology over the central Himalaya

**Jaydeep Singh[1], Narendra Singh[1*], Narendra Ojha[2], Amit Sharma[3], Andrea Pozzer[4, 5], Nadimpally Kiran Kumar[6], Kunjukrishnapillai Rajeev[6], Sachin S. Gunthe[7], V. Rao Kotamarthi[8]**

[1]Aryabhatta Research Institute of observational sciencES (ARIES), Nainital, India

[2]Physical Research Laboratory, Ahmedabad, India

[3]Department of Civil and Infrastructure Engineering, Indian Institute of Technology Jodhpur, Jodhpur, India

[4]Department of Atmospheric Chemistry, Max Planck Institute for Chemistry, Mainz, Germany

[5]Earth System Physics Section, International Centre for Theoretical Physics, Trieste, Italy

[6]Space Physics Laboratory, Vikram Sarabhai Space Centre, Thiruvananthapuram, India

[7]EWRE Division, Department of Civil Engineering, Indian Institute of Technology Madras, Chennai, India

[8]Environmental Science Division, Argonne National Laboratory, Argonne, Illinois, USA

**Correspondence:** Narendra Singh (narendra@aries.res.in) and Andrea Pozzer (andrea.pozzer@mpic.de)

**Abstract**

The sensitive ecosystem of the central Himalayan (CH) region, experiencing enhanced stress from anthropogenic forcing, requires adequate atmospheric observations and an improved representation of the Himalaya in the models. However, the accuracy of atmospheric models remains limited in this region due to highly complex mountainous topography. This article delineates the effects of spatial resolution on the modeled meteorology and dynamics over the CH by utilizing the Weather Research and Forecasting (WRF) model extensively evaluated against the Ganges Valley Aerosol Experiment (GVAX) observations during the summer monsoon. WRF simulation is performed over a domain (d01) encompassing northern

India at 15 km x 15 km resolution, and two nests: d02 (5 km x 5 km) and d03 (1 km x 1 km) centered over
CH with boundary conditions from respective parent domains. WRF simulations reveal higher variability
in meteorology, e.g. Relative Humidity (RH=70.3–96.1%), Wind speed (WS=1.1–4.2 ms$^{-1}$), as compared
to the ERA-Interim reanalysis (RH=80.0–85.0 %, and WS=1.2–2.3 ms$^{-1}$) over northern India owing to the
higher resolution. WRF simulated temporal evolution of meteorological variables, is found to agree with
the balloon-borne measurements with stronger correlations aloft (r = 0.44–0.92) than those in the lower
troposphere (r = 0.18–0.48). The model overestimates temperature (warm bias by 2.8$^{o}$C) and
underestimates RH (dry bias by 6.4%) at the surface in d01. Model results show a significant improvement
in d03 (P=827.6 hPa, T=19.8 $^{o}$C, RH=92.3%), closer to the GVAX observations (P=801.4 hPa, T=19.5
$^{o}$C, RH=94.7%). Interpolating the output from the coarser domains (d01, d02) to the altitude of the station
reduces the biases in pressure and temperature, however, suppresses the diurnal variations but highlighting
the importance of well-resolved terrain (d03). Temporal variations in near-surface P, T and RH are also
reproduced by WRF in d03 to an extent (r > 0.5). A sensitivity simulation incorporating the feedback from
the nested domain demonstrates the improvement in simulated P, T and RH over CH. Our study shows
that the WRF model set up at finer spatial resolution can significantly reduce the biases in simulated
meteorology, and such an improved representation of CH can be adopted through domain feedback into
regional-scale simulations. Interestingly, WRF simulates a dominant easterly wind component at 1 km x
1 km resolution (d03), which is missing in the coarse simulations; however, the frequency of south-
easterlies remains underestimated. Model simulation implementing a high resolution (3s) topography input
(SRTM) improved the prediction of wind directions; nevertheless, further improvements are required to
better reproduce the observed local-scale dynamics over the CH.
**1. Introduction**
The Himalayan region is one of the most complex and fragile geographical systems in the world, and has
paramount importance for the climatic implications and air composition at regional to global scales (e.g.

Lawrence et al., 2010, Pant et al., 2018; Lelieveld et al., 2018). The ground-based observations of meteorology and fine-scale dynamics are highly sparse and limited. In this direction, an intensive field campaign known as the GVAX (Kotamarthi, 2013) was carried out over a mountainous site in the central Himalaya which provided valuable meteorological observations for atmospheric research, model evaluation and further improvements. Accurate simulations of meteorology are needed for numerous investigations, such as to study the regional and global climate change, snow-cover change, trapping and transport of regional pollution, and the hydrological cycle, especially the monsoon system (e. g. Sharma and Ganju, 2000; Bhutiyani et al., 2007; Pant et al., 2018). Studies focussing over this region have become more important due to the increasing anthropogenic influences resulting in enhanced levels of Short-Lived Climate forcing Pollutants (SLCPs) along the Himalayan foothills (e. g. Ojha et al., 2012; Sarangi et al., 2014; Rupakheti et al., 2017; Deep et al., 2019; Ojha et al., 2019). Although Global Climate Models (GCMs) simulate the climate variabilities over global scale, their application for reproducing observations in the regions of complex landscapes is limited, due to coarse horizontal resolution (e. g. Wilby et al., 1999; Boyle et al. 2010; Tselioudis et al., 2012; Pervez and Henebry, 2014; Meher et al., 2017). Mountain ridges, rapidly changing land-cover, and the low altitude valleys often lie within a grid box of typical global climate models resulting in significant biases in model results when compared with observations (e. g. Ojha et al., 2012; Tiwari et al., 2017, Pant et al., 2018). On the other hand, Regional Climate Models (RCMs) at finer resolutions allow better representation of the topographical features, thus providing improved simulations of the atmospheric variability over regions of complex terrain. Several mesoscale models (e. g. Christensen et al., 1996; Caya and Laprise 1999; Skamarock et al., 2008; Zadra et al., 2008) have been developed and applied successfully over different parts of the world. These studies have revealed that the RCMs provide significantly new insights by parameterizing or explicitly simulating atmospheric processes over finer spatial scales. Nevertheless, large uncertainties are still seen over highly complex areas indicating the effects of further unresolved terrain features (e. g. Wang et al., 2004; Laprise, 2008; Foley, 2010) and need to improve the simulations.

Anthropogenic influences and climate forcing have been increasing over the Himalaya and its foothill
regions since pre-industrial times (Bonasoni et al., 2012; Srivastava et al., 2014; Kumar et al., 2018).
Consequently, an increase in the intensity and frequency of extreme weather events has been observed
over the Himalayan region (e. g. Nandargi and Dhar, 2012; Sun et al., 2017; Dimri et al., 2017) in the past
few decades. These events include extreme rainfall and resulting flash floods, cloudbursts, landslides etc.,
and the associated weather systems range from mesoscale to synoptic-scale phenomena. Unfortunately,
the lack of an observational network covering the Himalaya and foothills with sufficient spatio-temporal
density inhibits the detailed understanding of the aforementioned processes, meteorological and dynamical
conditions in the region. Therefore, usage of regional models, evaluated against available in-situ
measurements can fill the gap of investigating atmospheric variability in the observationally sparse and
geographically complex mountain terrain of the Himalaya.
The biases in simulating the meteorological parameters especially in the lower troposphere are associated
with several factors, e.g. representation of topography, land use, surface heat and moisture flux transport,
and parameterization of physical processes (e. g. Lee et al., 1989; Hann and Yang, 2001; Cheng and
Steenburgh, 2005; Singh et al., 2016). The WRF has been used for the model experiments over complex
terrain around the world, e.g., the Himalaya region (e.g. Sarangi et al., 2014; Singh et al., 2016, Mues et
al., 2018; Potter et al., 2018; Norris et al., 2020; Wang et al., 2020), Tibetan Plateau (e.g. Gao et al., 2015;
Zhou et al., 2018), and the multiple mountain ranges in the western United States (e.g. Zhang et al., 2013)
to evaluate and study the meteorology and dynamics. A cold bias was reported in this model over the
Tibetan Plateau and the Himalayan region by Gao et al (2015). The near-surface winds showed biases
linked with unresolved processes in the model such as sub-grid turbulence, land-surface atmospheric
interactions, besides boundary layer parametrization (Hanna and Yang, 2001; Zhang and Zheng, 2004;
Cheng and Steenburgh, 2005). Zhou et al (2018) found lower biases in simulated winds after considering
the turbulent orographic formed drag over the Tibetan Plateau.
The WRF model with suitably chosen schemes has been shown to reproduce the regional-scale
meteorology (Kumar et al., 2012) and to some extent also the mountain-valley wind systems (Sarangi et
al., 2014) and boundary layer dynamics (Singh et al., 2016; Mues et al., 2018) over the Himalayan region.
Nevertheless, local meteorology is still difficult to simulate accurately; Mues et al (2018) performed high-
resolution WRF simulation over the Kathmandu valley of Himalaya and reported overestimation of 2m
temperature and 10m wind speed, which they attributed to insufficient resolution of the complex
topography, even at a resolution of 3 km. Although few studies have used the WRF model at very high
resolution over the Himalayan region (e.g., Cannon et al., 2017; Mues et al., 2018; Potter et al., 2018;
Zhou et al., 2018; Zhou et al., 2019; Norris et al., 2020; Wang et al., 2020), still the model performance
over the complex terrains like the Himalaya requires improvement which can be done through an extensive
evaluation at sub-kilometer resolution against an intensive field campaign. The main objectives of the
study are as follows:
1. To examine the model performance over the CH at varying resolutions (15km, 5km and 1km) by

111         evaluating several model diagnostics against the observations made during the GVAX campaign.

2. To investigate the effect of feedback from nest to the parent domain, as this might allow

113         configuring a model setup covering the larger Indian region with more accurate results over the

114         Himalaya

3. The downscaling to a sub-kilometer (333m) resolution with the implementation of a very high

116         resolution (3 s) topographical input into the model to examine the potential of simulations in

117         reproducing local-scale dynamics

The subsequent section 2 describes the model set up, followed by experimental design, and a discussion
of datasets used for model evaluation. Section 3 provides a comparison of model results with the ERA-
Interim reanalysis (section 3.1), radiosonde observations (section 3.2), and ground-based measurements
(section 3.3). Analysis of domain feedback is presented in section 3.4, and the effect of implementing
high-resolution topography is investigated in section 3.5, followed by the summary and conclusions in
section 4.

## 2. Methodology

### 2.1 Model set up and Experimental Design

The WRF model–version 3.8.1 has been used in the present study. WRF is a mesoscale non-hydrostatic,
Numerical Weather Prediction (NWP) model with advance physics and numerical schemes for simulating
meteorology and dynamics. WRF-ARW uses an Eulerian mass-based dynamical core with terrain-
following vertical coordinates (Skamarock et al., 2008). ERA-Interim reanalysis from the European Center
for Medium-Range Weather Forecasts (ECMWF) available at a temporal resolution of 6 hour and
horizontal resolution of $0.75^0$ x $0.75^0$ with 37 vertical levels from the surface to the top at 1 hPa (Dee et
al., 2011), has been used to provide the initial and lateral boundary conditions to the WRF model. Static
geographical data from Moderate Resolution Imaging Spectroradiometer (MODIS) available at 30s
horizontal resolution, is utilized for land use, and land cover.
The Goddard scheme is used for shortwave radiation (Chou and Suarez, 1994), while the longwave
radiation is simulated by the Rapid Radiative Transfer Model scheme (Mlawer et al., 1997). For resolving
the boundary layer processes the first order non-local closure based Yonsei University (YSU) scheme
(Hong et al., 2006) is used including an explicit entrainment layer with the K-profile in an unstable mixed
layer. PBL height is determined from the Richardson number ($Ri_b$) method in this PBL scheme.
Convection is parameterized by the Kain-Fritsch (KF) cumulus parameterization (CP) scheme, accounting
for sub-grid level processes in the model such as precipitation, latent heat release and vertical redistribution
of heat and moisture as a result of convection (Kain, 2004). With the increase in model grid resolution to
less than 10 km (known as "grey area"), the CP scheme is usually turned off, and cloud and precipitation
processes are resolved by the microphysics (MP) scheme (Weisman et al., 1997). In the present study, the
CP scheme is used for d01 while it is turned off for d02 and d03. The Thomson microphysics containing
prognostic equations for cloud water, rainwater, ice, snow, and graupel mixing ratios, is used (Thompson
et al., 2004). Parameterization of surface processes is done with MM5 Monin-Obukhov scheme and
Unified Noah land surface model (LSM) (Chen and Dudhia, 2001; Ek et al., 2003; Tewari et al., 2004).
The Noah LSM includes a single canopy layer and four soil layers at 0.1, 0.2, 0.6 and 1m within 2m of
depth (Ek et al., 2003).
The model is configured with three domains of 15 km (d01), 5 km (d02) and 1 km (d03) horizontal grid
spacing using Mercator projection centering at Manora Peak (79.46°N, 29.36°E, amsl ~ 1936m) in the
central Himalaya. The topography within the model domains is highly complex, as evident from the ridges
(Figure 1). The outer domain d01 includes the northern part of Thar Desert, part of Indo-Gangetic Plain
(IGP), and the Himalayan mountains, while the innermost domain d03 consists of mostly mountainous
terrain. The model has 51 atmospheric vertical levels with the top at 10 hPa. For d01, 100 east-west and
86 north-south grid points are used to account for the effect of synoptic-scale meteorology, e.g. Indian
summer monsoon. The d02 has 88 east-west and 76 north-south grid points covering sufficient spatial
region around the observational site to consider the effects of mesoscale dynamics, e.g. change of wind
pattern due to orography. The innermost domain, d03, has 126 east-west and 106 north-south grid points
to resolve local effects, e.g. convection, advection, turbulence, and orthographic lifting.

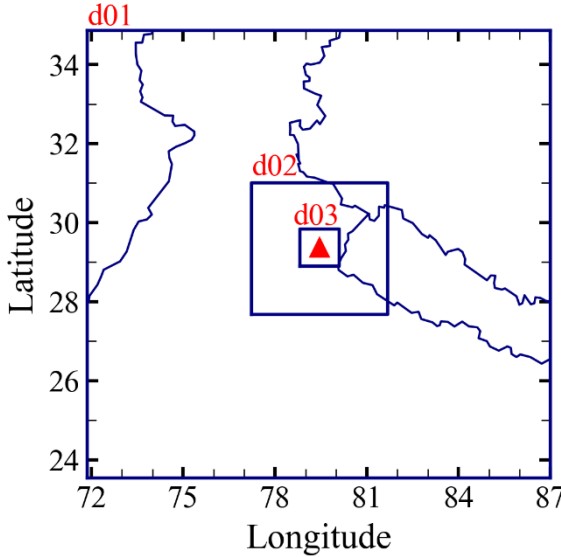


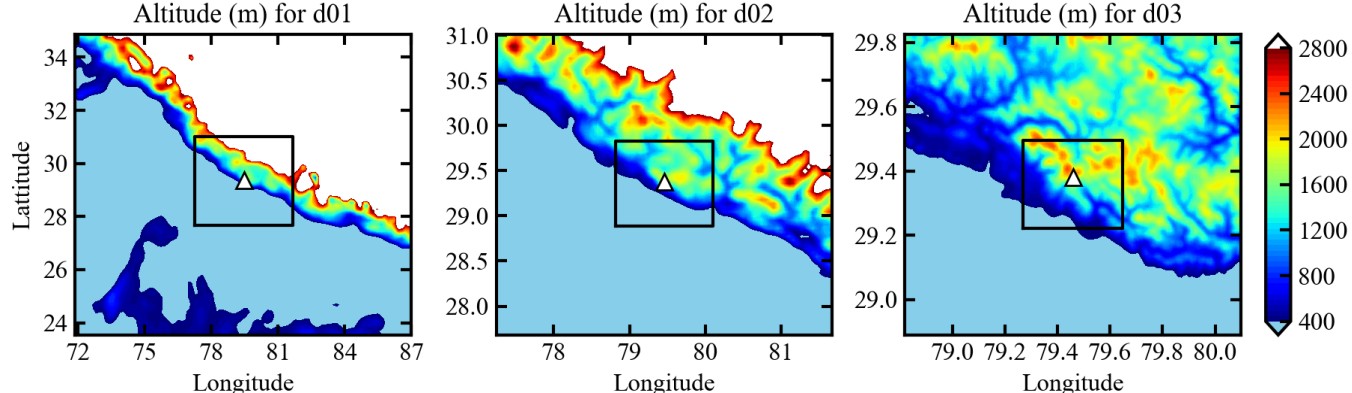


**Figure 1:** Topography represented in the WRF model domains with three horizontal resolutions, namely- domain d01 (15 x 15km), domain d02 (5 x 5km) and domain d03 (1 x 1 km). Each box inside corresponds to the nested domain (upper panel). Triangle in the innermost box indicates the location of the GVAX-campaign site i.e. Manora peak Nainital. Bottom panel belongs to the topography of each individual domain (left to right). The finest nest inside of the d03 (bottom right) is d04 at the resolution of 333m (discussed in section3.5)

For d01, boundary conditions are provided from ERA-Interim reanalysis, as explained earlier. Model simulations have been performed for the four months of summer monsoon: 1 June 2011 to 30 September 2011 (JJAS). This simulation period is chosen considering the availability of continuous observations from

11 June 2011 and to allow sufficient spin-up time of 10 days for the model to achieve its equilibrium state
(Angevine et al., 2014; Seck et al., 2015; Jerez et al., 2020). Only, the outer domain d01 is nudged with
the global reanalysis for temperature, water vapor, zonal and meridional (u and v) components of wind
using nudging coefficient of 0.0006 (6 x $10^{-4}$) at all vertical levels (e.g. Kumar et al., 2012). Several of the
configuration options e. g. physics and meteorological nudging are selected following earlier applications
of this model over this region (e. g. Kumar et al., 2012; Ojha et al., 2016; Singh et al., 2016; Sharma et al.,

179    2017).


**2.2. Observational data**
We utilize the observations conducted during an intensive field campaign- the Ganges Valleys Aerosol
Experiment (GVAX) to evaluate model simulations. The GVAX campaign was carried out using
Atmospheric Radiation Measurement (ARM) Climate Research Facility of the U.S. Department of Energy
(DOE) from 10 June 2011 to 31 March 2012 at ARIES, Manora Peak in Nainital (e.g. Kotamarthi, 2013;
Singh et al., 2016; Dumka et al., 2017). This observational site (79.46ºN, 29.36ºE, 1940m above sea level)
is located in the central Himalaya, as shown in Figure 1. The surface-based meteorological measurements
of ambient air temperature, pressure, relative humidity, precipitation, wind (speed and direction) were
made using an automatic weather station at 1-minute temporal resolution. The instantaneous values of the
observations are compared with hourly instantaneous model output at the nearest grid point.
The vertical profiles of temperature, pressure, relative humidity and horizontal wind (speed and direction)
were obtained by four launches (00:00, 06:00, 12:00 and 18:00 UTC) of the radiosonde each day during
the campaign (Naja et al., 2016). The continuous vertical profiles of the meteorological parameters except
wind speed and direction were available from the end of June 2011 to the entire study period, whereas
valid and quality wind data were available only for September 2011. Hence, in this study, radiosonde
measurements from 1 July 2011 onwards are used for the model evaluation of meteorological parameters,
except wind speed and direction, which are evaluated only for September. A total of 309 valid profiles of
temperature and relative humidity and 104 profiles of wind are used. The statistical metrics such as mean
bias (MB), root mean square error (RMSE) and correlation coefficient (r) are used for the model
evaluation, and the description of these metrics is given in the supplementary material.

**3.    Results and Discussions**
**3.1. Comparison with ERA-Interim reanalysis**
Here, we have used the ERA-Interim data for comparison with WRF output. We first compare the WRF
simulated spatial distribution of meteorological parameters (surface pressure, 2m air temperature, 2m RH
and 10m WS) with ERA-Interim reanalysis over the common region of all the domains and averaged for
the entire simulation period (Figure 2). The three contours of the topographic height of 500m, 1500m and
2000m are used to relate the meteorological features to the resolved topography in three domains. The
common area in all domains includes low-altitude IGP region in the south (elevation of less than 400m,
Figure 1) and elevated mountains of central Himalaya in the north. Also, for a consistent comparison,
model simulated values are taken at the same time intervals as that in ERA-Interim data (i.e. every 6h).
From the comparison presented in Figure 2, it is evident that the meteorological parameters simulated by
the model are dependent on the model grid resolution. The existence of the sharp gradient topographic
height (SGTH) of about 1600 m from the foothill of the Himalaya to the observational site modifies the
wind pattern as well as moisture content differently at different grid resolutions, indicating the critical role
of mountain orography. The surface pressure explicitly depends upon the elevation of a location from
mean sea level. The contour of the pressure parameter from ERA-Interim data shows the surface pressure
of about 900hPa for observational site Manora Peak and varied from 550 to 975 hPa within this region,
while WRF simulated pressure is 869 hPa, 835hPa and 827 hPa for d01, d02 and d03, respectively.  WRF
simulated surface pressure ranges from 821.9 hPa over high altitude CH region to 977.0 hPa in IGP region
within d01. Simultaneously, the range of variation in the surface pressure is 788.1 – 977.5 and 760.4–
977.7 hPa within d02 and d03 respectively, and the minimum pressure decreases from d01 to d03 which
is attributed to the improvement in resolved topography on increasing model grid resolution. However,
the effects of the SGTH are not observed for temperature, wind and RH in ERA-Interim contours, due to
the unresolved topographic features. Simulated maps show spatial homogeneity of meteorological
parameters over the flat terrain of IGP in the foothills of Himalaya as compared to elevated central
Himalayan region.

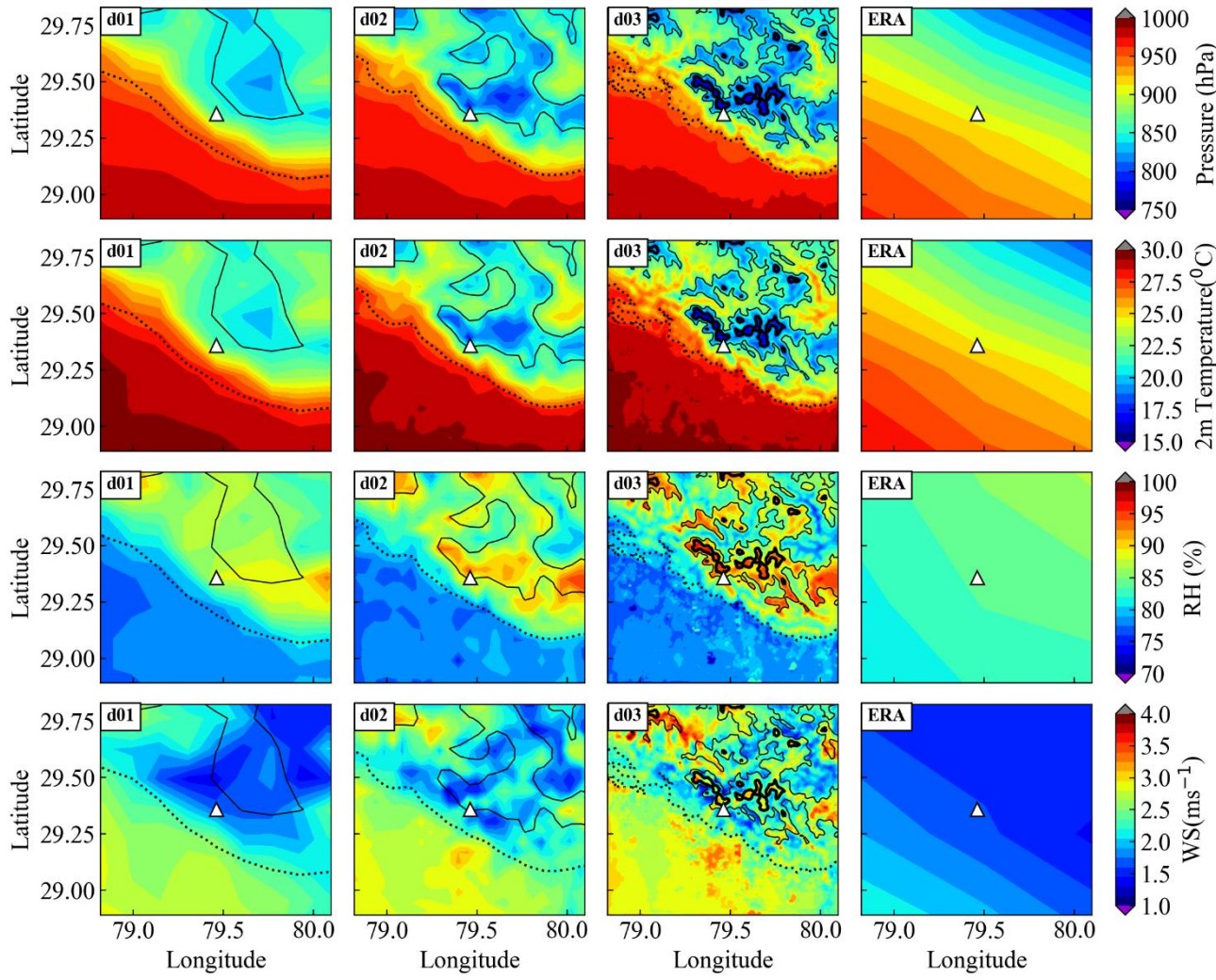


**Figure 2**. Contours in the first three columns show WRF results for the three domains (first column: d01,
second column: d02 and third column: d03) and the fourth column shows corresponding parameters from
the ERA-Interim reanalysis. First row shows mean surface pressure during the monsoon (JJAS), the

second row shows 2m temperature (in $^0$C), 3rd row shows 2m relative humidity (RH; %) and the bottom row shows 10 m wind speed (WS; ms$^{-1}$) along with three elevation contours at 500m (dashed), 1500m (thin solid), and 2000m (thick solid).

The effect of spatial resolution is clearly observed over the mountainous region of the Himalaya, where the size of the mountains changes abruptly, with the modelled output showing increasingly distinct features with increasing grid resolution. On the other hand, there are minimal differences in the topography of the IGP, and hence the meteorological features associated with the topography are well captured in the model even at a coarser resolution of 15 km.

Model simulations show the topography dependent spatial variation in 2m temperature in the ranges of 20.0–29.5$^0$C in d01, 17.3–29.6$^0$C in d02, and 15.5.0–29.9$^0$C in d03, with lowest values simulated over the elevated mountain peaks and higher values over the temperate IGP region. The contours in three model domains show an explicit dependency of 2m temperature on the grid resolution over the mountainous region. With the increasing model resolution, the topography is resolved to a greater extent, and lower temperature is simulated at higher surface elevations, as expected. Further, the estimation of water vapour is essentially needed for both climate and numerical weather prediction (NWP) applications. Figure 2 shows that the simulated relative humidity is above 70% in all three domains for the monsoon season. The variations (minimum-maximum) in the relative humidity in ERA-Interim (80% - 85%) data set, over the domains d01 (77-93%), d02 (74 – 95%), and d03 (70- 96%) are generally comparable. The mountain slopes provide the uplift to the moist monsoonal air that on ascent subsequently saturates and increases the relative humidity to about 90% as observed over the grid encompassing the site. Contour lines (1500m and 2000m in Figure 2) depict the low pressure and temperature with the higher relative humidity feature of the peaks and these features are sharper as the resolution increases from d01 to d03.

The wind speed is highly dependent upon the model grid resolution as well as orography-induced circulations during different seasons (Solanki et al., 2016; 2019), and reflected by Figure 2. As mentioned

earlier, although the topography of the IGP region does not vary abruptly, the magnitude of the wind speed
over this region as well as over the complex Himalayan region is found to change significantly at different
model resolutions, thereby, indicating that the wind speed is very sensitive to both model resolution and
topography. The wind speed in d01 varies from 1.3 ms$^{-1}$ to 2.8 ms$^{-1}$, while the wind variations in domains
d02 (1.2 – 3.4 ms$^{-1}$ ) and d03 (1.3 – 4.2 ms$^{-1}$) show higher variability than that in the ERA-interim (1.2–
2.3ms$^{-1}$) due to finer resolution of the WRF. Overall, the impact of the topography resolved at higher
resolution in WRF shows the contrasting differences in surface pressure, temperature, relative humidity
and wind speed compared to coarse resolution ERA-interim dataset.
**3.2. Comparison with Radiosonde observations**
The intensive radiosonde observations made during the GVAX field campaign at Manora Peak (79.46°N,
29.36°E, amsl ~ 1936m) in the central Himalaya (shown in Figure 1) are used for the evaluation of model
resolutions. The comparison of the model simulated profiles of temperature, relative humidity, and wind
speed against the radiosonde observations are shown in Figure 3.

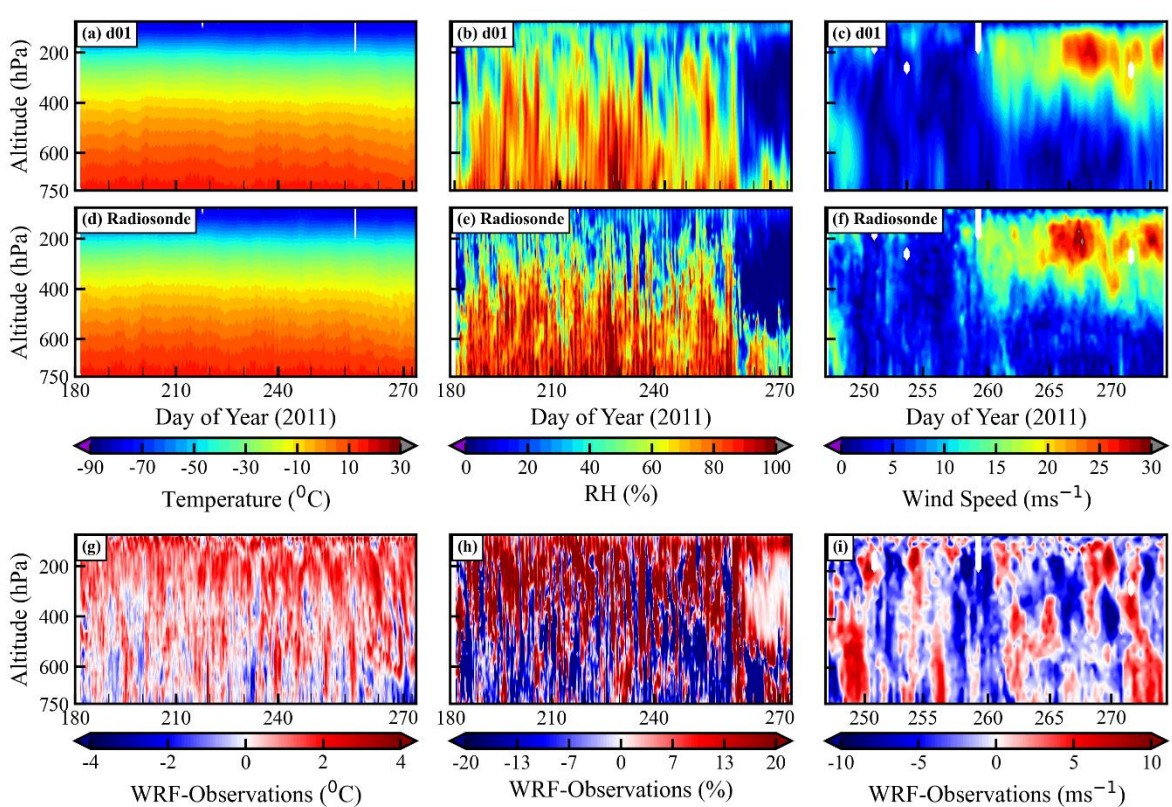


**Figure 3:** The comparison of simulated vertical profiles of **(a)** temperature ($^0$C), **(b)** relative humidity (RH; %), and **(c)** wind speed (ms$^{-1}$) in d01 with the radiosonde observations (d, e, and f). The x-axis of (a), (b), (d), (e), (g) and (h) show the day of the year 2011 starting from 1 July (182$^{nd}$ day) to 30 September (273$^{rd}$ day). The vertical profiles of wind speed (c, f) are plotted only for September 2011. The third row (g, h, and i) shows the difference in temperature, relative humidity, and wind speed, between WRF d01 simulation and radiosonde observation.

The inversion of temperature at the top of the troposphere occurred at ~90hPa (~16km) in observations (Figure 3d, Figure S1) whereas, radiosonde profiles show that temperature decreases with pressure from 15.5$^0$C at 750 hPa to -78.0 $^0$C at ~90 hPa. As evident from the simulated temperature profiles, the WRF model well captured these features and found to show a reduction from 15.1 $^0$C to -76.6 $^0$C in these pressure levels. Further, the differences between model (d01) and radiosonde observations (Figure 3g) range from -4 to 4$^0$C. The mean RH values from the radiosonde observations (model d01) also show decrease from 82.3% (76.7%) at 750 to 25.2% (32.0%) at 90 hPa. The mean RH difference between observation and model (Figure 3h) shows that the model simulates more humid atmosphere at higher altitudes while showing a low-humidity bias in lower altitudes. The wind data from radiosonde measurements available for September 2011, were utilized to compare the model output. Observations and modelled winds are ≤10 ms$^{-1}$, within the altitude region of the surface to about 400hPa (~7 km) till mid of September (day of the year 258). Wind increases (≥15 ms$^{-1}$) above 400 hPa and attains maximum values (≥ 25 ms$^{-1}$) between 250 and 100hPa after 258 day of the year (15$^{th}$ September 2011). However, simulated winds are slightly lower and less wide spreads as compared to observations. The comparison of the wind profiles with the same x-axis (shown in Figure S2) with other meteorological parameters show that the lower relative humidity (<30%) is observed along with higher wind speed during this period. In general, the vertical profiles and variation of simulated wind speed agree well with the observation. The Taylor diagram (Taylor, 2001), in Figure 7a, is used to express the statistical comparison between model simulations and observations. In the diagram, the comparison is summarized with correlation coefficient (r), normalized

root mean squared difference (RMSD) and standard deviation, normalized by that of the observations (SD). In most cases, the model simulates less variability in meteorological parameters as shown by the normalized standard deviation, which is less than 1. For temperature and wind speed, model shows good correlation (r) with the observations at 250hPa ($r > 0.80$) than that in lower altitudes i.e. 750hPa ($r < 0.40$). On the other hand, the model captures variability in humidity relatively well at 500 hPa ($r = 0.71$) but shows poor correlation at 75 hPa ($r = 0.17$) near the model top.

Lower correlations for temperature and wind speed near to the surface (750 hPa) could be due to the terrain induced effects which are most significant in the local boundary layer. The surface-level winds and turbulence are some of the boundary layer features, affected mainly by the surface and terrain characteristics. The vertical profiles of these parameters up to 500 hPa in all three model domains are shown in Figure 4. Differences between the simulated vertical profile of temperature and radiosonde observation are in general similar in all the domains. Except for the relative humidity in d01, other meteorological parameters (temperature and wind speed) do not reveal strong dependencies on the model resolution. However, the model overestimates the relative humidity near 500hPa level in d02 and d03 on some of the days. In case of the wind speed, the model underestimates the magnitude of the wind in the first few days up to 500 hPa, though by and large the model is able to capture the vertical profiles.

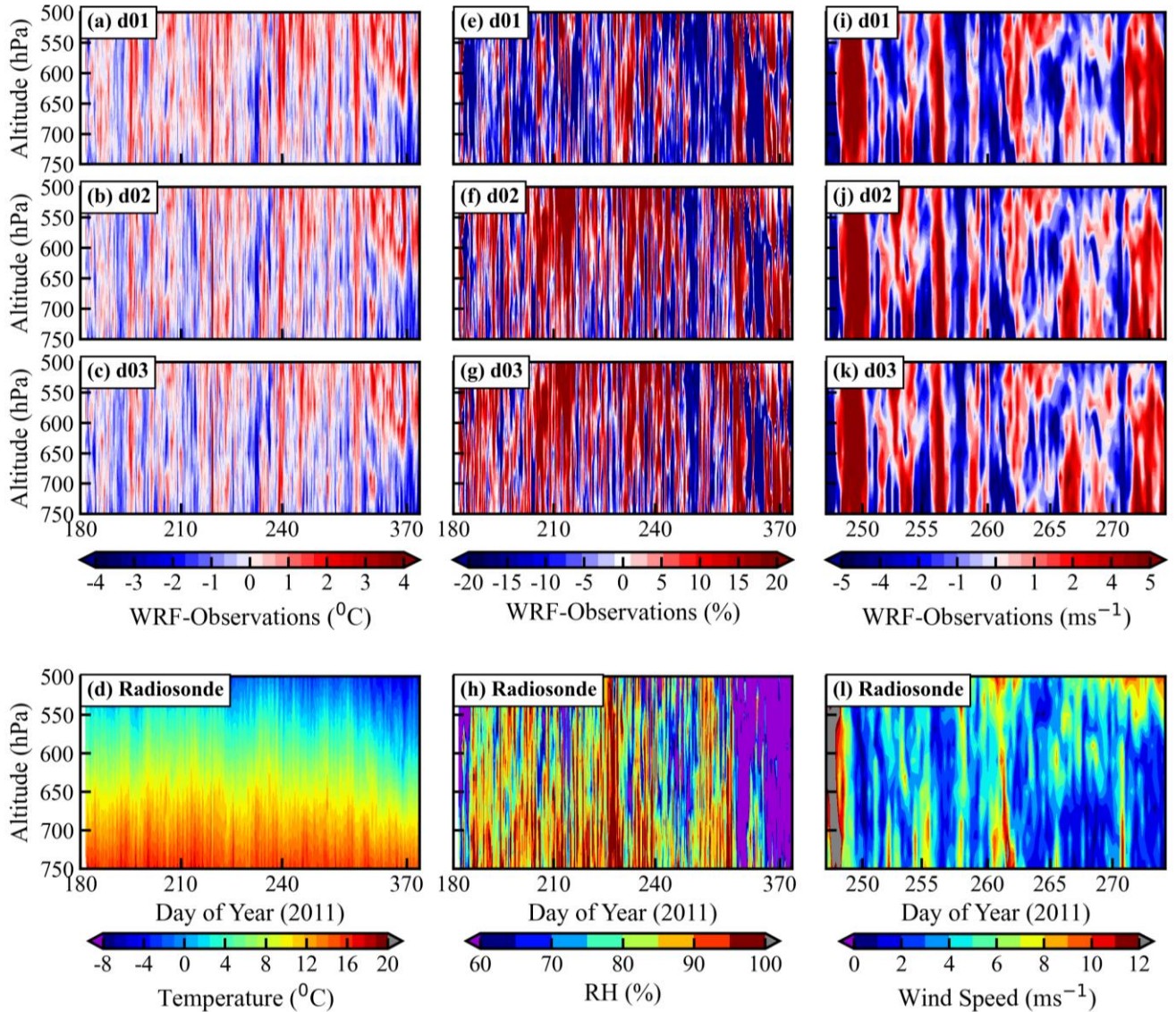

**Figure 4:** Difference between model (d01: first row, d02: second row, d03: third row) and radiosonde observation for temperature (first column: a, b, c), relative humidity (second column: e, f, g) and wind speed (third column: i, j, k) profiles up to 500 hPa. The fourth row provides the vertical profiles of radiosonde measurements. The x-axis of the panels (a-h) shows the day of the year 2011 from 1 July (182nd day) to 30 September (273rd day). Wind speed ($ms^{-1}$) profiles in the panels (i-l) are provided for September 2011.

Figure 5 shows the vertical profiles of the statistical metrics: mean bias (MB), root mean square error (RMSE), and correlation coefficient (r) for temperature, relative humidity, and wind speed for the three simulations (d01, d02, d03). The magnitudes of the MB values throughout the troposphere are estimated

to be within about 1 $^{\circ}$C, 12 %, and 2.5 ms$^{-1}$ for temperature, relative humidity, and wind speed,
respectively. Additionally, RMSE values are about 1 $^{\circ}$C, 15–30%, and 2.5–5 ms$^{-1}$ for temperature, relative
humidity, and wind speed, respectively. As discussed earlier, correlations between model results and
observations are found to be stronger in the middle and upper troposphere than in the lower troposphere.
For temperature, the r values are higher than 0.75 between 600 to 200 hPa, whereas it decreases up to 0.4
at lower altitudes, i.e. near 800 hPa. Correlations in the lower troposphere are notably weaker (r = ~0.25)
in case of wind speed. The results suggest that the model captures well the day-to-day variabilities in the
meteorological parameters in the middle and upper troposphere and to a minor extent in the lower
troposphere. Relatively weaker correlations in the lower troposphere are suggested to be associated with
more pronounced effects of the uncertainties caused by the underlying complex mountain terrain and
resulting unresolved local effects. Wind fields near the surface are strongly impacted by interactions
between terrain and boundary layer besides orographic drag in a modelling study over the Tibetan Plateau
(Zhou et al., 2018) and in measurements over the Himalaya (Solanki et al., 2019). Increase in bias with
altitude was reported by Kumar et al (2012) for dew point temperature. Besides the model physics, the
higher uncertainties in radiosonde humidity observations might have also contributed to these differences.
The effect of model resolution is not very significant for temperature and wind profiles above 800 hPa;
nevertheless, the mean bias for RH is lower (~5%) in the 800–600 hPa altitude range and higher in the
450-300 hPa altitude range in the d02 and d03 simulations. This might be arising due to deep convection
in the model at a higher resolution. Overall, the model captured the vertical structures of meteorological
parameters; however, better representation of complex terrain itself is insufficient for improving the model
performance aloft. On top of the, better representation of topography as considered here, it highlights the
need for future studies evaluating various physics scheme. Nevertheless, model biases have been
significantly reduced for surface level meteorology with higher resolution, and the details are discussed in
the subsequent section.

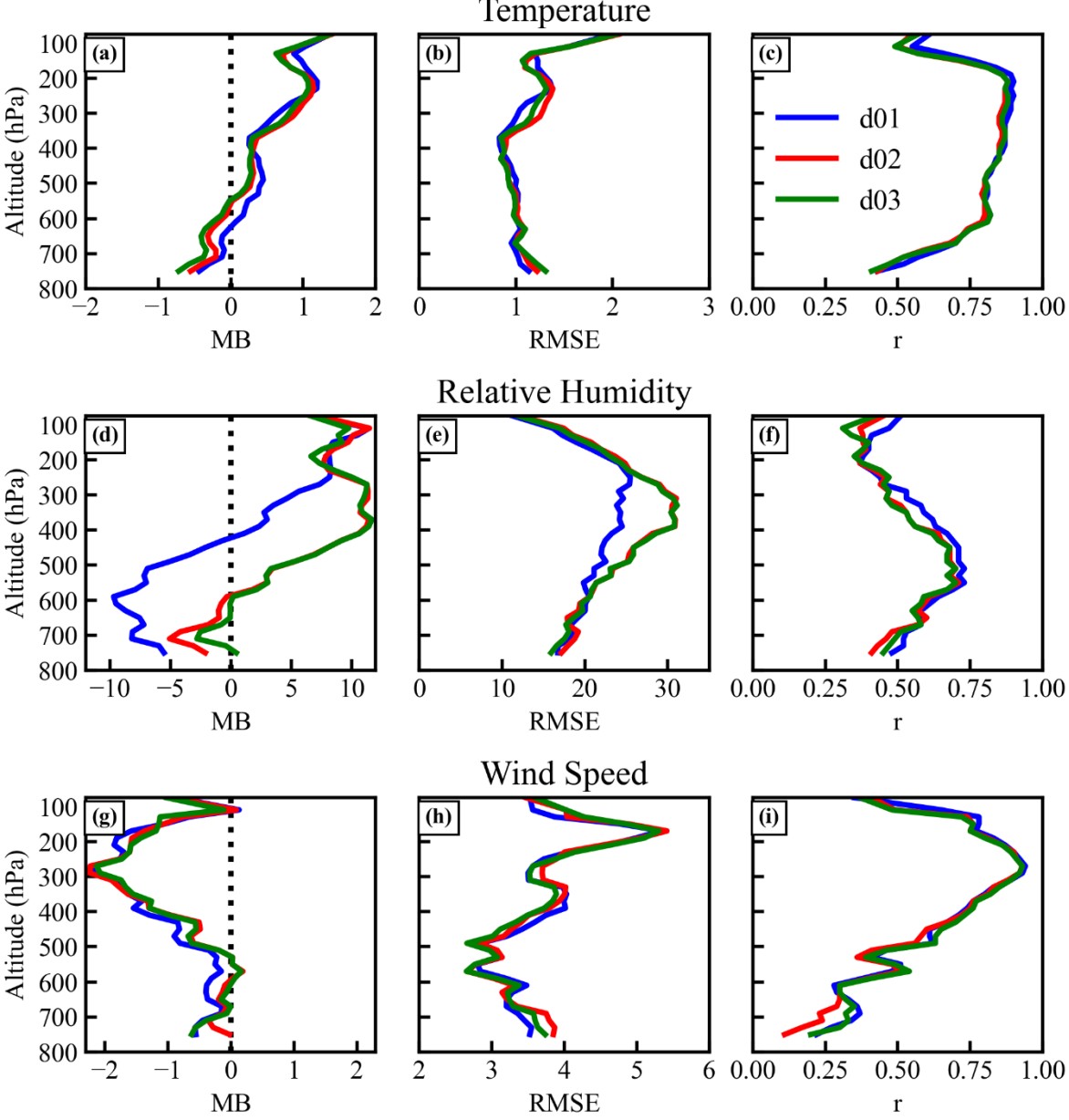

**Figure 5.** The vertical profiles of mean bias (MB), root mean square error (RMSE), and correlation coefficient (r) for temperature, relative humidity, and wind speed for different domains d01 (blue), d02 (red) and d03 (green).

## 3.3. Comparison with ground-based observations

The model simulated 2m temperature (T2), 2m relative humidity (RH2) and 10 m wind speed (WS10) for the observational site, Manora Peak are compared with the ground-based measurements made during

GVAX campaign in Figure 6 and summarized in Table1. The diurnal variations in T2, RH2 and WS10
simulated by the WRF model are compared with observations, whereas the surface pressure does not show
a significant diurnal variation (not shown here). Model simulation d01 shows a positive bias of 68 hPa in
surface pressure with strong correlation (r = 0.97) with observation (mean = ~801 hPa). A significant
improvement is achieved (MB = 26 hPa) in d03 as a result of the finest resolution simulation (Figure 6
and Table 1). The WRF model simulated T2 shows warm bias in all three domains. The simulated T2 for
d01 varies from 16.2 to 28.7 $^0$C with the higher mean value of 22.3±2.1$^0$C as compared to the observed
mean value of 19.5±1.6$^0$C with a correlation of r = 0.75 between d01 and observation. This warm bias is
seen to decrease from d01 (2.8 $^0$C) with the increasing model resolution to 0.2$^0$C in d03 simulation (Table
S1). The mean value of the RH2 in d01 is about 88.2±9.7%, 6.4% lower than the observed value 94.7±9.5%
with the correlation about 0.45 (Figure 7b). MB and RMSE values of RH2 show a decrease with increasing
model resolution (Table S1). As relative humidity also depends on temperature; therefore, the diurnal
variation in 2m specific humidity (Q2; g kg$^{-1}$) has also been analyzed (Figure S3). Q2 is observed in the
range of 5.5–21.5 g kg$^{-1}$ with the mean value as 16.8±2.0 g kg$^{-1}$. It is found that the agreement in Q2 is
relatively better (MB = -0.7 g kg$^{-1}$; r = 0.77 in d03), when the statistical metrics are compared with that
for RH2 (Table S1). The wind speed plays a vital role in transport processes and controls the dynamics of
the atmosphere at different temporal and spatial scales. The average 10m wind speed (WS10) during
monsoon over the measurement station is about 2.1±1.4 ms$^{-1}$ which is quite comparable to that simulated
in d01 (2.1±1.1 ms$^{-1}$) whereas overestimated in d02 by 0.9 ms$^{-1}$ and in d03 by 0.5 ms$^{-1}$ (Table 1 and Table
S1). In the case of the WS10, the correlation is 0.18 for d01 and d02, which improves to 0.24 in d03. The
diurnal variation of WS10 (Figure 6c) is not well captured, especially during the noontime.
**Table 1:** Mean (± Standard deviation) along with the minimum and maximum values of the meteorological
parameters: surface pressure (P; hPa), 2m Temperature (T2; $^0$C); 2m relative humidity (RH2; %) and 10m
wind speed (WS10; ms$^{-1}$) in the model simulations and observations for the full observation period. An

additional evaluation is presented, accounting for the difference in model surface altitude and actual altitude of measurements (referred to as 'With altitude adjustment').

| Parameter | Without altitude adjustment | | | With altitude adjustment | | | Observation |
|---|---|---|---|---|---|---|---|
| | **d01** | **d02** | **d03** | **d01** | **d02** | **d03** | |
| P (hPa) | 869.6±2.6 | 835.3±2.5 | 827.6±2.4 | 801.3±2.4.6 | 801.3±2.4 | 801.4±2.4 | 801.1±2.4 |
| Min/Max | 862.8/875.1 | 828.3/840.8 | 821.2/833.1 | 795.0/806.7 | 795.0/806.7 | 795.2/806.8 | 795.1/806.8 |
| T2 ($^0$C) | 22.3±1.8 | 20.4±1.8 | 19.8±1.1 | 18.4±0.8 | 18.4±0.9 | 18.3±0.9 | 19.5±1.1 |
| Min/Max | 16.2/28.7 | 15.1/26.0 | 14.0/25.0 | 16.1/20.9 | 15.5/21.8 | 15.6/22.1 | 14.8/25.6 |
| RH2(%) | 88.2±9.7 | 94.3±6.4 | 92.3±7.9 | 86.2±10.9 | 93.8±8.5 | 91.5±9.7 | 94.7±9.5 |
| Min/Max | 53.3/100 | 67.6/100 | 52.3/100 | 43.9/100 | 51.3/100 | 47.9/100 | 31.6/100 |
| WS10 (ms$^{-1}$) | 2.1±1.1 | 3.0±1.4 | 2.6±1.7 | 3.4±2.6 | 4.8±3.1 | 4.0±3.1 | 2.1±1.4 |
| Min/Max | 0.0/8.6 | 0.1/11.4 | 0.1/11.7 | 0.0/20.2 | 0.1/23.9 | 0.1/22.1 | 0.0/10.0 |

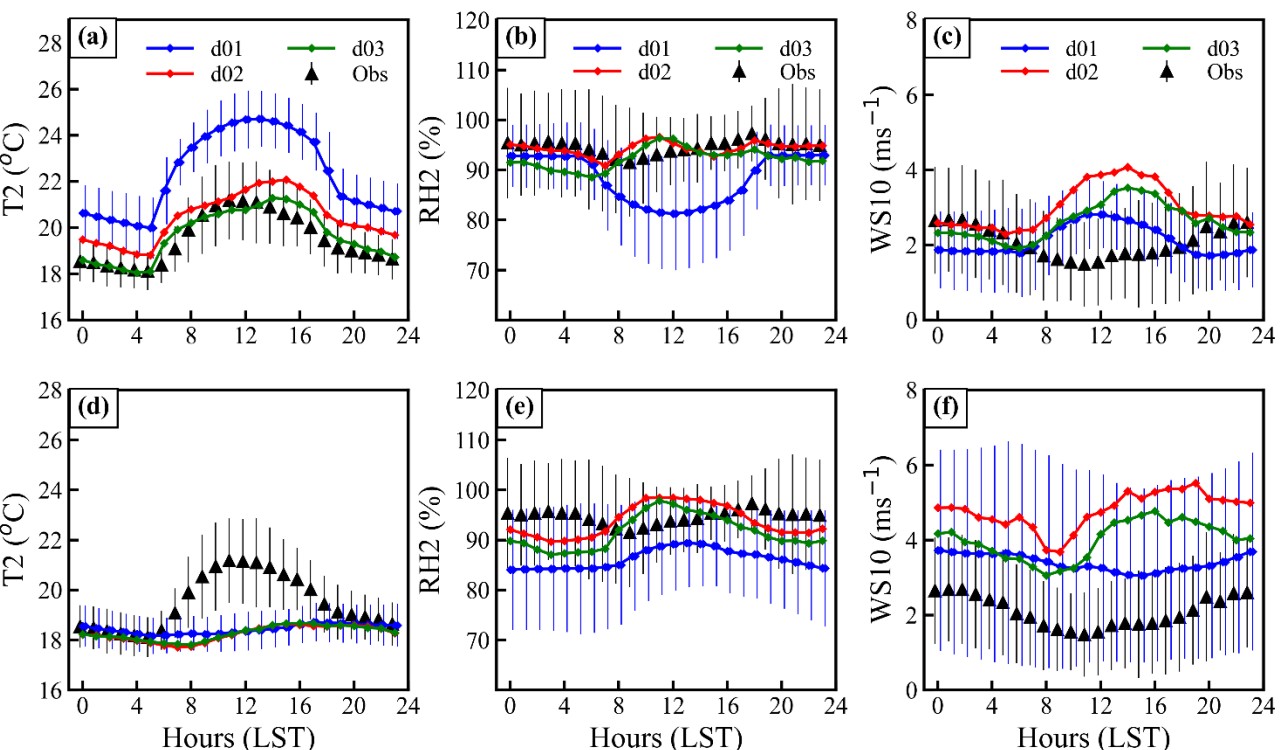

**Figure 6:** Mean diurnal variations of **(a)** 2m temperature: T2, **(b)** 2m relative humidity: RH2, and **(c)** 10m wind speed: WS10 from model simulations (d01, d02 and d03) and observations. Altitude adjusted

variations are also shown (d–f). The bars represent the standard deviation and shown only for domain d01
and the observations in order to avoid the overlap.

Due to the complex terrain and the grid size of the model, the simulated altitude of the observational site
could differ from reality. In this study, the model underestimated station altitude by about 588m, 480m,
and 270m in d01, d02, and d03. We performed an additional evaluation to explore and achieve the possible
improvement linearly interpolating the vertical profile of meteorological parameters to the actual altitude
of the station (Figure d-f), as done in a few previous study (e.g. Mues et al., 2018). The altitude adjustment
has been as per the equation of linear interpolation given in the supplementary material (equation - 4). The
analysis shows that the correlation coefficient values between model and observations do not show any
clear improvement (e.g. for T2 correlation coefficient is 0.35) on adjusting the altitude in model output
except for WS10. After adjusting the altitude, the temperature variability is suppressed by the model at
diurnal (Figure 6 a, d) and day to day timescales, i.e. r drops from 0.67 to 0.36 in d03. The comparison of
temperature among the three domains for additional model layers has also been analyzed (Figure S4). The
diurnal amplitudes are seen to be smaller at higher model layers. Additionally, the differences among
different simulations (d01, d02, and d03) also decrease at higher layers. As expected, the altitude
adjustment does reduce bias in pressure. Nevertheless, reductions in mean biases are not achieved (Table
S1); instead, absolute values of biases increase from 0.2 to 1.2, 2.4 to 3.1, 0.5 to 1.9, 0.7 to 1.6 in T2, RH2,
WS10, and Q2 in simulation d03. Besides thermal and mechanical interactions of the mountain surfaces
with atmosphere, local processes such as evaporation and transpiration affect the near-surface
meteorological conditions. A reduction in wind speed during the daytime is associated with the competing
effects of the mountain-valley circulations due to heating of the slopes versus synoptic-scale flows
(Solanki et al., 2019). To resolve such sub-grid scale processes, we emphasize that very high-resolution
simulations are needed as conducted in this study in order to simulate the meteorological variability in a
satisfactory way. The analysis further highlights a need of accurate representation of the complex
topographical features rather than altitude adjusted estimations which led to very limited improvements in
this case. However, we will discuss the evaluation without altitude adjustment until stated otherwise.
We evaluate the MB values (Table 1 and Table S1) in model simulations considering the benchmarks as
suggested by Emery et al (2001). In the d03 simulation, MB values for both T2 ($0.2^0$C) and Q2 (-0.7 g kg$^-$
$^1$) are found to be well within the range of benchmark values: $\pm0.5^0$C for T2 and $\pm1.0$ g kg$^{-1}$ for Q2. It is
important to note that biases in T2 in the coarser simulations d01 ($2.8^0$C) and d02 ($0.9^0$C) are however
higher as compared to the benchmarks. MB values in T2 estimated, for this representative Himalayan site,
are found to be slightly lower (+0.2 $^0$C) (-1.2 $^0$C with altitude adjustment) than these over the Tibetan
Plateau (-2 to -5 $^0$C) (Gao et al., 2015) and over mountainous regions in the Europe (Zhang et al., 2013).
Warmer bias in our case is due to underestimation of the Himalayan altitude, whereas, model
overestimated terrain height over the Tibetan Plateau region giving contrasting results. Further, the RMSE
in wind speed is lower (1.6–2.0 ms$^{-1}$) than that over Kathmandu valley (2.2 ms$^{-1}$; Mues et al., 2018) and
similar to benchmark (2.0 ms$^{-1}$). Mar et al (2016) also reported akin bias (2 ms$^{-1}$) in the 10m wind speed
over Europe and the average correlation of ~0.4–0.6 over the Alps.  Nevertheless, simulating the diurnal
variation of near-surface winds still remains challenging over complex terrain, but the bias was reduced
after including effects of the turbulent orographic form drag (Zhou et al., 2017; 2019). Besides turbulent
orographic form drag, it is suggested that wind speed is sensitive towards boundary layer schemes (Yver
et al., 2013; Zhou et al., 2019) and that more studies are needed to explore these aspects over the central
Himalaya.

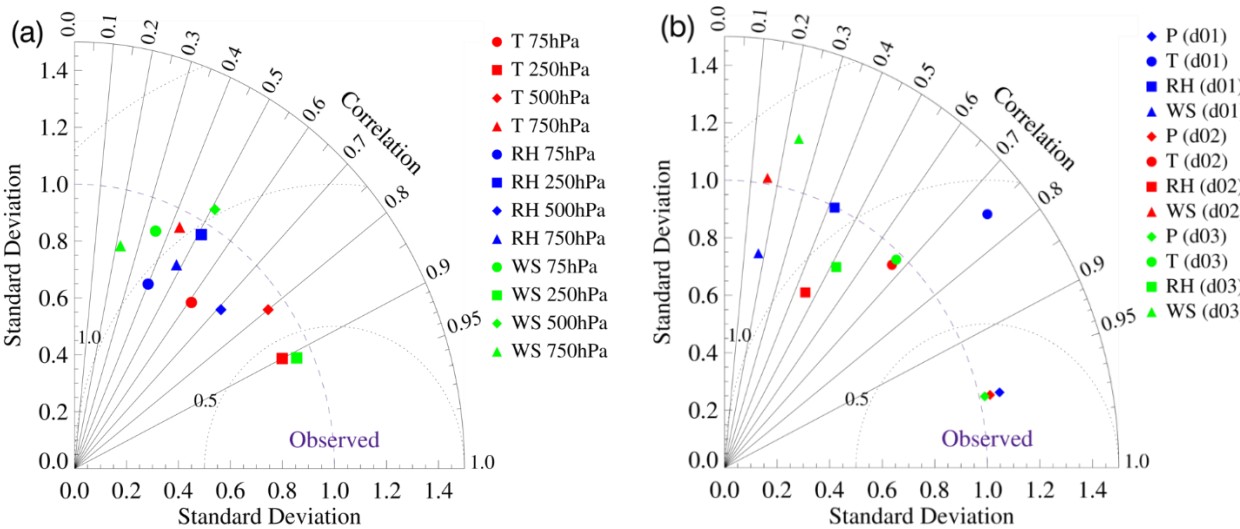


**Figure 7:** Taylor diagram with the correlation coefficient, normalized standard deviation, and normalized root mean square difference (RMSD) error for (a) model performance at different pressure levels shown in Figure 3 for d01, and (b) the model simulated surface pressure, 2m temperature, RH and 10 m wind speed for different domains as shown in Figure 6a-c.

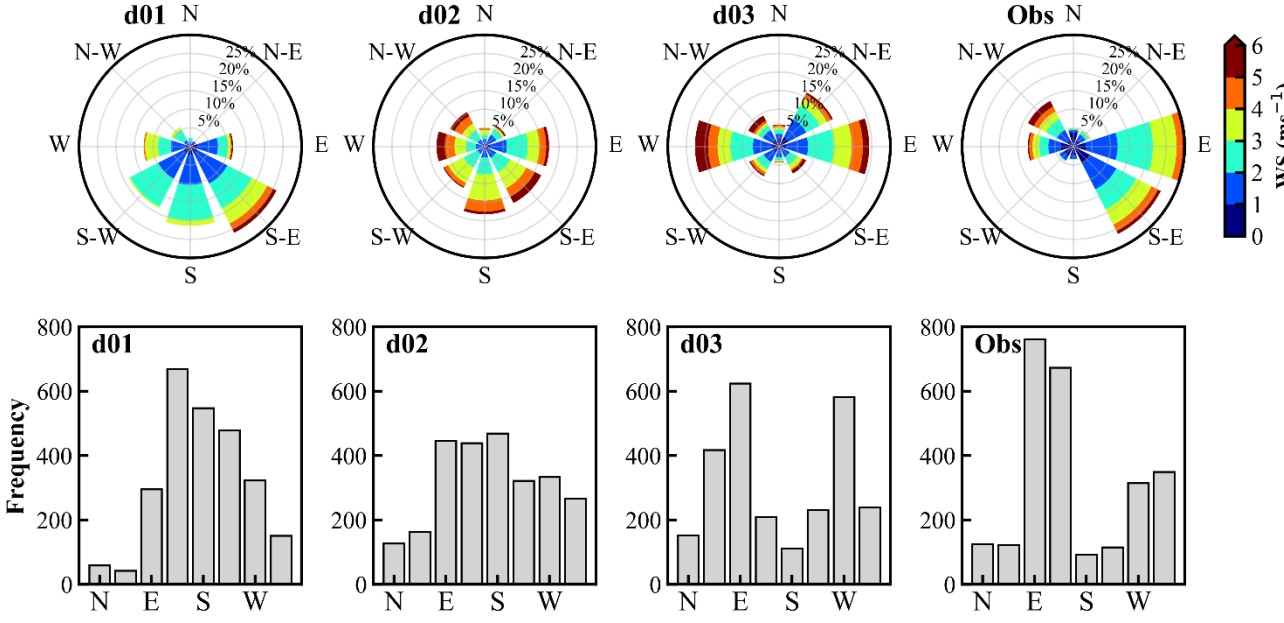

433

434

**Figure 8:** Comparison of the wind speed and direction represented by the wind rose (top panel), and frequency distribution of wind direction (bottom panel) for model simulations over the three domains (d01, d02, d03) and observations (obs) during June-September 2011. Different colours and radii of wind roses show the wind speed and frequency of occurrences, respectively.

The wind direction is strongly influenced by the surrounding topography over the mountainous region and the evaluation of the wind direction at horizontal resolution is depicted in Figure 8. The winds varying between meteorological direction $337.5^0$ and $22.5^0$ are considered to be the Northerly and represented by N in the frequency distribution and so on for other directional flow, referring to the clockwise meteorological convention. The wind flow dominance over the observational site is easterly (30%) and south-easterly (26%) while 26% occurrences of wind are from the west and north-west. The percentage of southerly (21%) and south-westerly (SW, 19%) are relatively higher in d01 as compared to the observations (S: 4% and SW: 5% respectively) and decreases to 4% and 9% in d03. Model is able to simulate the northerly and north-easterly winds in d01 and d02, while, the model simulates larger contribution of north-easterly and westerly winds in d03 which is not witnessed by the observations. The easterly component of the model simulated wind shows better agreement with observations on increasing the model resolution. In addition, the model is able to simulate the westerly and north-westerly wind contribution in d02, whereas, the westerly component is overpredicted by 10 percentage points in d03. The observations show that winds blowing from north, north-easterly, south and south-westerly are very weak (<2 ms$^{-1}$), and amount to be about ~15% of the total occurrences. The diurnal variation of the wind direction is not investigated here, however the impact of mountain topography on the near surface flow under low wind conditions has been discussed elsewhere (Solanki et al., 2019). Overall, the simulated wind field in d03 is relatively in better agreement with observations than d01 and d02. This is further assessed in section 3.5 using a finer resolution simulation by implementing SRTM 3s terrain data.

**3.4. Effect of feedback**

In the preceding section, the simulations were carried out without any feedback (WRF-WF) from the finer resolution domain to its parent domain, and results have been discussed. This WRF-WF experiment was conducted in such a way that it could explicitly account for the grid resolution effects on the model performance. The simulated meteorology with this model setup (with feedback) depicted different model performance in outermost coarse resolution domain d01 compared to d02 and d03. The model performance depends upon the boundary and initial conditions. Another model simulation is carried out in this section using the same configuration but with two-way interactive nesting and feedback (WRF-F) from nested domain to its parent domain. The simulated meteorological parameters in the higher nests, are fed back to its parent domains, and the boundary conditions replaced there. The model results over the CH region, in the regional scale simulation (d01), shows better agreement with the observations, because of the feedback from high resolution nested simulation. The comparison of the simulated meteorological parameters (T2, RH2, and WS10) for the outermost domain with the surface observations is presented in Figure 9 for both WRF-WF and WRF-F simulations thus showing the effect of the feedback within the outermost domain.

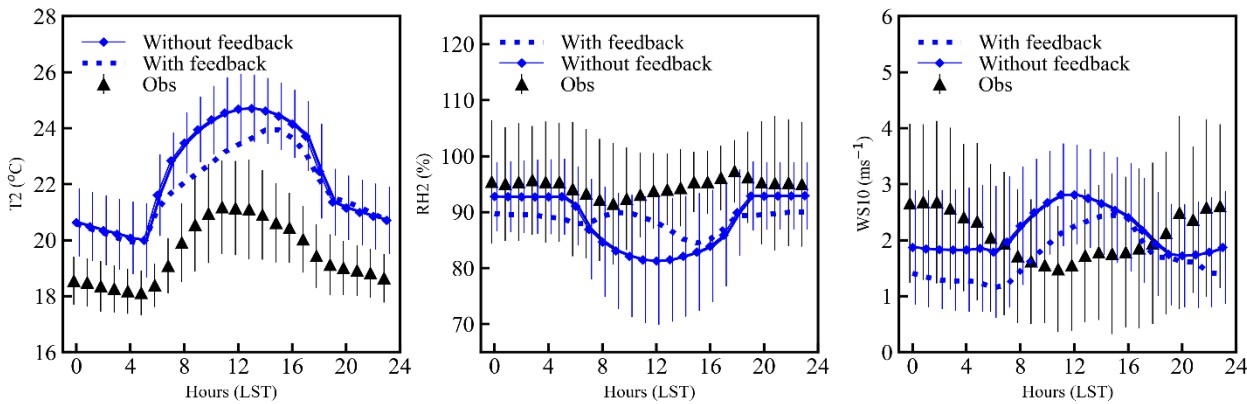

**Figure 9:** Diurnal variation of the T2, RH2, and WS10 from d01 without feedback (WRF-WF) and with feedback (WRF-F) simulations. The bars represent the standard deviation and shown only for domain d01 and observations in order to avoid the overlap.

The comparison of mean values (Table 2) shows a decrease in model bias for T2, RH2, and Q2 by 0.5 $^{0}$C, 0.3%, and 0.2 g kg$^{-1}$ respectively due to feedback from finer resolution simulations. Additionally,

correlations are found to show improvements for RH2 and Q2 by 0.15 and 0.12, respectively, due to
feedback, hence, the diurnal variation of relative humidity is more closer to the observation (Figure 9).
Nevertheless, smaller changes were seen in correlations for WS10 (by 0.05) and T2 (by -0.02) (Figure S5).
Variations in wind speed and direction also show significant improvements, especially in the dominant
flow direction, e.g. easterly, westerly and north-westerly (Figure S6).

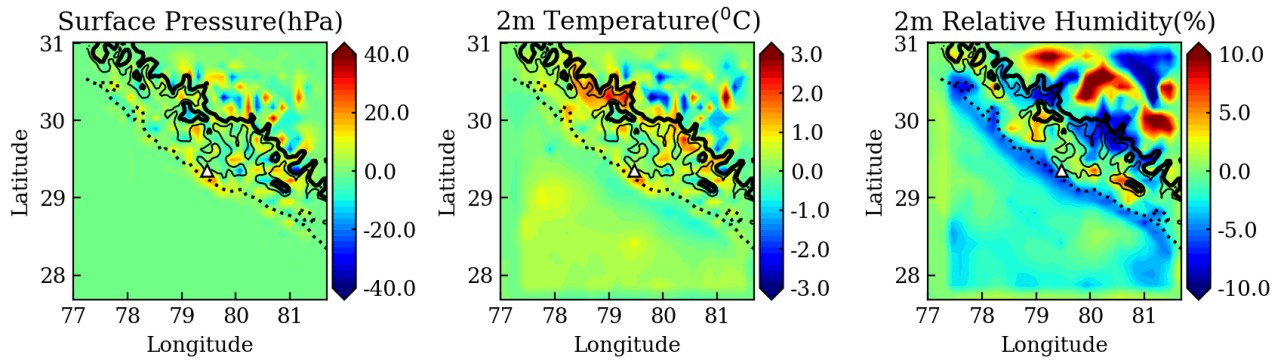


**Figure 10:** The effect of the two-way nesting on d01 is shown. The difference between the simulations
with feedback (WRF-F) and without feedback (WRF-WF) is shown for surface pressure, 2m temperature,
and 2m relative humidity along with three elevation contours at 500m (dashed), 1500m (thin solid), and
2000m (thick solid).
**Table 2:** Comparison of the simulated meteorology for surface pressure (P), 2m Temperature (T2), 2m
relative humidity (RH2), 10m wind speed (WS10) and 2m specific humidity (Q2) in two model
simulations: WRF-WF and WRF-F in the outermost domain d01  to the observations.

| Parameters | Observed | WRF-WF | WRF-F |
|:---:|:---:|:---:|:---:|
| P (hPa) | 801.4±2.4 | 869.6±2.6 | 858.9±2.5 |
| T2 ($^0$C) | 19.5±1.6 | 22.3±2.1 | 21.9±1.4 |
| RH2 (%) | 94.7±9.5 | 88.2±4.9 | 88.6±4.9 |
| WS10 (ms$^{-1}$) | 2.1±1.4 | 2.1±1.1 | 1.7±1.3 |
| Q2(g kg$^{-1}$) | 16.8±2.0 | 17.3±2.0 | 17.0±2.1 |


Effects of the feedback on surface pressure, 2m temperature and relative humidity in the domain d01 are shown by Figure 10. Feedback effects are seen to be more pronounced over the mountainous region than over the flat terrain of the IGP. The feedback from the nested domain to the parent domain mostly modifies the meteorology over the mountainous region, as shown by the topography contours in Figure 10. The analyses of biases and correlations suggest an improvement in the model simulated pressure, temperature, and humidity through feedback from well-resolved nests. This further underpins that better representations of the Himalaya over local-scales can be adopted to simulate meteorology at the regional-scale with lower biases over complex terrain in the given domain. Nevertheless, further modelling studies alongside with more observations are needed to improve the model performance. We extended the efforts to improve the wind speed and direction simulated over the complex topography by implementing a high resolution (3s) topographical input in the model as to evaluate finer resolution features over the Himalaya in the following section.

**3.5. Inclusion of high resolution (3s) SRTM topography**

Simulations described in previous sections were performed using the 30s (~1km) topographic data from the GMTED2010 (Danielson and Gesch, 2011) which is comparable to the highest resolution of the WRF simulation (d03). In order to evaluate the influences of topographical features on the wind flow at finer scales, the topography input available at very high resolution (3s or ~90m) from the Shuttle Radar Topography Mission (SRTM3s) (Farr et al., 2007) have been utilized. However, retaining the model configuration same as earlier, an additional innermost nest d04 having a resolution of ~333m, as depicted in Figure 1 (bottom right panel) is included. Simulation with SRTM data at 1 km resolution did not differ significantly with the similar resolution simulation using GMTED2010 (GMTED hereafter). For this experiment, model simulation is performed only for September 2011. This simulation is carried out without feedback and compared with the observation to check the effect of implementing high-resolution topography.

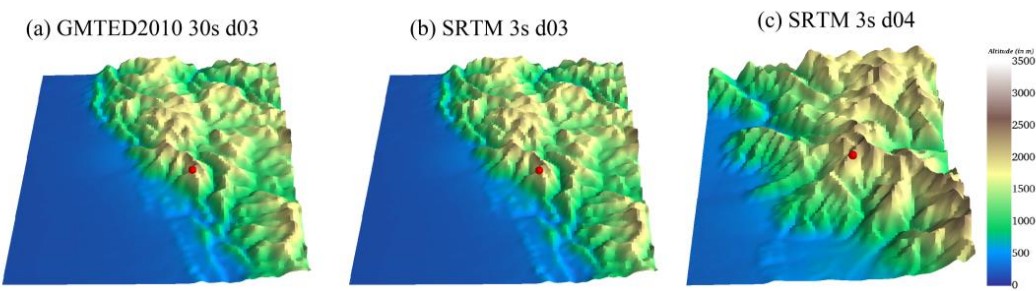

Red dot reperests observatinal site Manora Peak


**Figure 11:** The topography from GMTED at 30s in domain d03 (a), SRTM at 3s in domain d03 (b) and

SRTM at 3s in domain d04 (c). The model elevations of the observational site in d03 and d04 are 1670m
and 1876m respectively.

The comparison of the topographic height between GMTED and SRTM3s, in Figure S7, shows that the
differences are larger over the mountainous region, which vary from -100 to +100m. The differences are
suppressed within d02 and d03 as the topography input is changed from GMTED to SRTM3s datasets.
The topography in d04 (Figure 11c) gets resolved better, and marked by sharp variations of mountain
ridges and valleys using the SRTM3s as compared to d03, which are smoothed out with GMTED (Figure
11a) or with SRTM3s interpolated to 1 km (Figure 11b). After including the SRTM3s topography, the MB
value for wind speed in d03 is found to show a slight reduction (~0.04 ms$^{-1}$). Also, the flow from various
directions exhibited an improvement of 1-2% with the use of SRTM3s (Table S2).
In d04, surface pressure is seen to be simulated more realistically (809 hPa), and the dry bias in 2m relative
humidity is improved by ~2% (Figure S8 and Table S2). Simulations of diurnal wind variations remain
challenging (not shown here) even at finest resolutions considered (d04), even utilizing the updated
topographic data (SRTM3s). Further, to understand the effect of SRTM3s data in d04, the wind direction
is compared with that in d03 and observations.  The variations in winds are analysed and shown by the
wind rose (Figure 12a-d) and frequency distribution (Figure 12e-h). The fraction of north-easterly
component in d04 with SRTM3s (5%) is found to be comparable with observations (6%), which was
overestimated by19% (17%) in d03 with GMTED (SRTM3s). The frequency of the southerlies improves

with the increasing resolution of topography and matches better with the observations. The observations show the prevalence of north-westerly (19%), easterly (24%), westerly (18%) and south-easterly (20%) winds and these are also seen to be dominant directions in the simulation d04 with the exception of south-easterly winds.

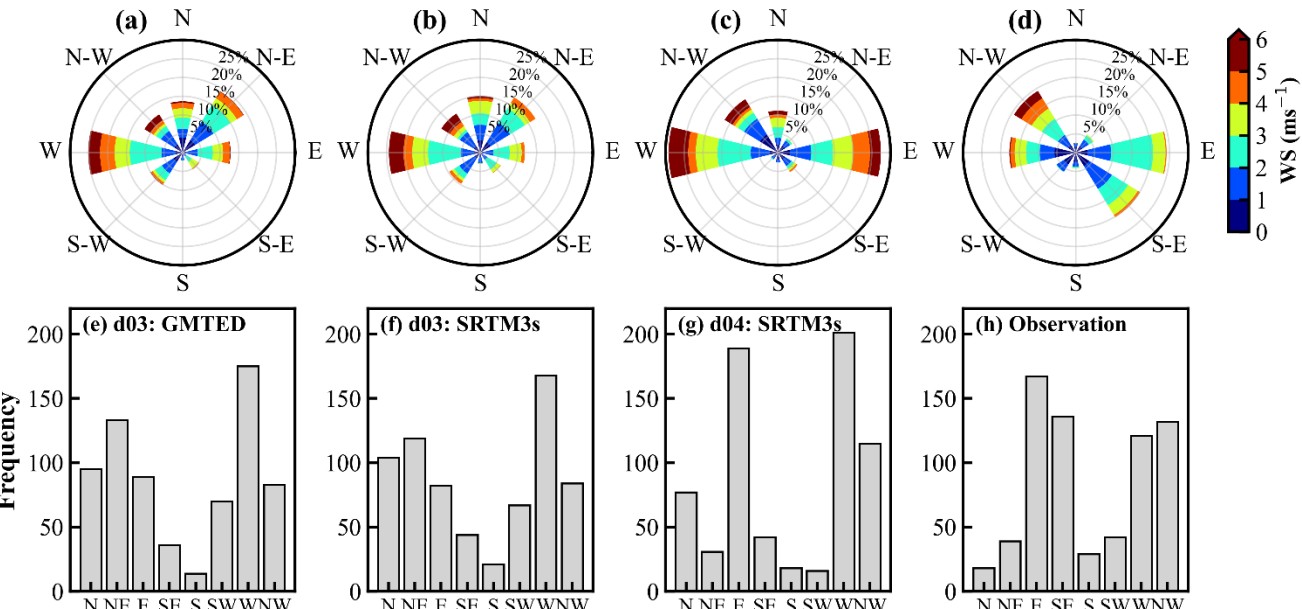

**Figure 12:** Wind roses (a) d03 using GMTED, (b-c) d03 and d04 using SRTM3s topography data, and (d) surface observation. Corresponding frequency distribution of the wind directions are shown from e-f. The comparison of the wind speed and direction are shown for the month of September, 2011.

Simulation of the wind directions improved from d03 to d04 by using the SRTM3s topography, except certain wind directions such as south-easterly. An improvement is noticed in simulated surface pressure, 2 m relative humidity and 10 m wind speed using the SRTM3s topography. Topographical data at different resolutions is found to show the RH differences in the range of -1 to 1% in d02 and -3 to 3% in d03 (Figure S9). Differences in simulated RH could be associated with the multi-scale orographic variations which are found to be the key factors in meteorological simulation over complex terrain (e.g. Wang et al., 2020). The effects of the SRTM3s topographic static data have been studied previously over other regions of the world (e. g. Teixeira et al., 2014; De Meij and Vinuesa, 2014). However, the daytime lower wind speed

and the transition phases during morning and evening hours still remain a challenge even after using the
high resolution (333m x 333m) nest. Such discrepancies between model and observations over the
Himalayan region are suggested to be associated with still unresolved terrain features, besides the
influences of input meteorological fields as well as the model physics on simulated atmospheric flows (e.
g. Xue et al., 2014; Vincent et al., 2015).

**4. Summary and Conclusions**
This study using WRF-model mainly elucidated upon the various diagnostics it calculates for its multiple
domains, the comparison of model results to an intensive field campaign and downscaling to a sub-
kilometer resolution with 3s resolution SRTM topography data that resolves individual peaks and valleys
over the CH region. The effects of spatial resolution on model simulated meteorology have examined by
combining the WRF model with ground-based, and in-situ observations, and reanalysis datasets. Owing
to the highly complex topography of the central Himalaya, model results show strong sensitivity towards
the model resolution and adequate representation of terrain features. Model simulated meteorological
profiles do not show much dependency on the resolution, except in the lower atmosphere, which is directly
influenced by terrain induced effects and surface characteristics emphasizing the need to evaluate various
physics schemes over this region. The biases in 2 m temperature, relative humidity and surface pressure
show a decrease on increasing the model resolution indicating a better resolved representation of
topographical features. Diurnal variations in meteorological parameters also show better agreements on
increasing the grid resolution. Although the surface pressure does not show a pronounced diurnal variation,
nevertheless, the biases in simulated surface pressure reduce significantly over fine-resolution simulations.
Interpolation of coarser simulations (d01, d02) to the station altitude reduces the bias in surface pressure
and temperature, but suppresses the diurnal variability. The results highlight the significance of accurately
representing terrains at finer resolutions (d03). Model is generally not able to reproduce the frequency
distribution of wind direction, except in some of the major components in all the simulations with varying
resolutions. The directionality of the simulated winds shows improvements over finer grid resolutions;
however, reproducing the diurnal variability still remains a challenge. Biases are stronger typically during
daytime and also during transitions of low to high wind conditions and vice versa. This is attributed to the
uncertainties in representing the interaction of slope winds with the synoptic mean flow and local
circulations, despite an improved representation of terrain features. A sensitivity experiment with domain
feedback turned ON shows that the feedback process can improve the representation of the CH in the
simulation covering a larger region of the northern Indian subcontinent. It is suggested that further
improvements in the model performance are limited due to the lack of high-resolution topographical input
biases through input meteorological fields, and model physics. Nevertheless, the implementation of a very
high resolution (3s) topographical input using the SRTM data shows the potential to reduce the biases
related to topographical features to some extent.

**Code and data availability**
Observational data from the GVAX campaign is available freely
(https://adc.arm.gov/discovery/#v/results/s/fsite::pgh.M). WRF is an open-source and publicly available
model, which can be downloaded at http://www2.mmm.ucar.edu/wrf/users/download/get_source.html. .
A zip file containing a) namelists for both pre-processor (WPS) as well as the WRF, b) 3s resolution
topography input prepared for the pre-processor, along with a README file describing the necessary
details to perform the simulations, has been archived at https://doi.org/10.5281/zenodo.3978569.

**Author contributions**
NS and AP designed and supervised the study. JS performed the simulations, assisted by NO and AS. JS,
NO, AS analysed the model results and NVPKK, KR, SSG contributed to the interpretations. VRK
contributed significantly in conceiving and realizing the GVAX campaign. JS and NS wrote the first draft,
and all the authors contributed to the manuscript.

**Acknowledgement**
This study has been supported by ABLN& C: NOBLE project under ISRO-GBP. We are thankful to
Director ARIES, Nainital. We acknowledge NCAR for the WRF-ARW model, ECMWF for the ERA-
Interim reanalysis data sets, ARM Climate Research Facility of U.S. Department of Energy (DOE) for the
observations made during the GVAX campaign. Computing resources from the Max Planck Computing
and Data Facility (MPCDF) are profoundly acknowledged. N. Ojha acknowledges the computing
resources - Vikram-100 HPC at Physical Research Laboratory (PRL), and valuable support from
Duggirala Pallamraju and Anil Bhardwaj. Constructive comments and suggestions from the anonymous
reviewers and the handling editor are gratefully acknowledged.

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
