# Peer review of "Effects of spatial resolution on WRF v3.8.1 simulated meteorology over the central"

_Geoscientific Model Development, 2020_

## Referee Comment (RC1) · Anonymous Referee #1 · 4 Apr 2020

This article delineates the effects of spatial resolution on the model performance over the central Himalaya. Ground and radiosonde profiles were used to assess the performance of WRF at different spatial resolution. The temporal evolution of meteorological profiles in WRF is seen to be in agreement with the measurements with stronger correlations for upper troposphere than those in the lower troposphere. To use the profiles to assess the model result for mountain region is new in my review. However, I find that this paper does not really reach to main question for mountain meteorology studies. The authors should review the frontier of this area. Only do evaluations is not qualified for GMDD publication. There are some problems with the figures which are not well arranged in a well-know way. An issue is that when they compare model grid values with that of AWS, they might use two temperature at different height. Please compare

the AWS elevation and the grid elevation where AWS located. Use the elevation difference to adjust the model temperature. The same problem also happens to wind speed. There are many evaluation papers for the mountain numerical simulation. The authors should review these papers, try to improve the wind speed performance.

Figure 3, add their difference between d01 and Radiosonde and give some introduction on the difference. Line 261, it's better to add a figure which shows the correlation coefficient r, mean bias etc result for all the height, not only say model captures variations at 500 hPa better than 50 hPa. Its also possible to compare the r and mean bias profiles with the three spatial resolution simulation. Figure 4, many things are not clear in the figure, which year? It also repeat with figure 3. Again, the difference is more interesting to us. Figure 6 the figure legend is not clear at all. Replot the figure with a colored marker. Figure 7 where is (a) and (b) letters? what does "0-6-12-..30" mean in the first wind-rose diagram? then why 0-2, 2-4, 4-6.....legend appears on the right of the fourth diagram? Figure 8, the simulation does not show the diurnal variation in wind speed at all. What's the explanation for it? This is really interesting for mountain numerical simulation.

[Figure]

---

## Referee Comment (RC2) · Anonymous Referee #2 · 6 Jul 2020

This study use WRF v3.8.1 to explore the effects of spatial resolution on local meteorology. It is very interesting that they found the finer spatial resolution can reduce the biases in simulated meteorology and improve representation of CH through domain feedback into regional-scale simulations. However, in this study, there are too many descriptions of the simulation, but no enough physical explanation to the simulation. It's difficult to make sense that why it occurred. In my view, this manuscript still needs major revision before it can be accepted.

section 2.2: How do you process the different temporal resolution of datasets, using the mean value or instantaneous value?

Line 182-184: It is available of ERA interim at $0.125° \times 0.125°$, but it's the interpolation results, which may not represent the true performance of ERA interim, especially over

the complex terrain regions. It's better to add the comparison between WRF and ERA interim at 0.75° × 0.75, even there is much less grids of ERA in D03.

Please update the figure captions: i.e., units of all the variables in Figure 3; caption of Figure 6 is not clear (Fig 6a is the comparison between WRF simulation at D01 and the observation?) ; Figure 8 is only focused D01, etc. you should make them clear in figure caption.

Line 259-262: Why did it happen? The different vertical distribution and the lower correlation at lower altitudes mostly come from the influence of land-air interaction. Please discussing the possible factors of your results.

How do you process the different spatial representation of different simulation and observation? For D01, one grid can indicate the mean situation of 15*15km area; meanwhile, for D02, it only indicates that in 5*5km area, etc. please show details of your methods to compare the grid simulation and the in-situ observation.

Line 286-287: It's very interesting that WRF shows a warm bias south side of Himalaya. Many previous studies pointed that there is obvious cold bias over Tibet (including Himalaya), i.e., Zhou et al. (2017) and Gao et al. (2015). Did you check your location of observation site and WRF grids? The warm bias in your WRF simulation is due to the lower terrain height of the grids than the Observed, please check if they are located over valley and the observed located over ridge

Figure 5: as the WRF resolution increasing, the diurnal cycle simulation of T and RH are better, but it didn't work for wind speed. please check the location of the WRF grids and observed station, if both them located valley or ridge? Besides, Zhou et al. (2019) stressed the importance of turbulent orographic form drag (TOFD) on the diurnal cycle simulation of wind speed. It's better to give more explains of inconsistent diurnal cycle of wind between simulation and observations.

RH is also dependent on Temperature. What's the performance of the WRF in simulation Specific Humidity (Q) ? Please compare the Q between WRF simulation and observation.

Section 3.4: What are the effects of feedback on the wind direction? In WRF-WF experiments, there are obvious difference among the simulated wind direction at three resolution. Is there any improvement in the WRF-F experiments?

Section 3.5: You should check the orographic variation in WRF model output, when you input different geographic data. multi-scale orographic variations are key factors of Wind and moisture simulation over complex terrain, i.e., south side of Himalaya (Wang et al., 2020)

References: Zhou, X., Yang, K., Beljaars, A. et al. Dynamical impact of parameterized turbulent orographic form drag on the simulation of winter precipitation over the western Tibetan Plateau. Clim Dyn 53, 707–720 (2019). https://doi.org/10.1007/s00382-019-04628-0

Gao YH, Xu JW, Chen DL (2015) Evaluation of WRF mesoscale climate simulations over the Tibetan Plateau during 1979–2011. J Clim 28(7):2823–2841. https://doi.org/10.1175/Jcli-D-14-00300.1

Wang, Y., Yang, K., Zhou, X. et al. Synergy of orographic drag parameterization and high resolution greatly reduces biases of WRF-simulated precipitation in central Himalaya. Clim Dyn 54, 1729–1740 (2020). https://doi.org/10.1007/s00382-019-05080-w

---

## Author Comment (AC1) · 12 Sep 2020

Response is attached in the form of supplement.

Please also note the supplement to this comment:
https://gmd.copernicus.org/preprints/gmd-2020-12/gmd-2020-12-AC1-supplement.zip

---

## Author Comment (AC2) · 12 Sep 2020

Response is attached in the form of supplement.

Please also note the supplement to this comment: https://gmd.copernicus.org/preprints/gmd-2020-12/gmd-2020-12-AC2-supplement.zip

---

## Author Response (AR1)

**Response to Reviewer #1**

Authors would like to thank the anonymous reviewer for thorough evaluation of our manuscript and constructive comments. Point-by-point responses to the reviewer's comments are given below in bold fonts and corresponding changes in the manuscript have been highlighted in red color.

**General comment:** This article delineates the effects of spatial resolution on the model performance over the central Himalaya. Ground and radiosonde profiles were used to assess the performance of WRF at different spatial resolution. The temporal evolution of meteorological profiles in WRF is seen to be in agreement with the measurements with stronger correlations for upper troposphere than those in the lower troposphere. To use the profiles to assess the model result for mountain region is new in my review. However, I find that this paper does not really reach to main question for mountain meteorology studies. The authors should review the frontier of this area. Only do evaluations is not qualified for GMDD publication.

Response: We agree with the reviewer and following the suggestion more literature survey has been included in the revised version (Page: 4-5, Lines: 85-104). We would like to mention that our study is not limited to evaluation only and we show that high-resolution set ups, with existing terrains in the model preprocessor, could reduce the model biases only to some extent. We therefore implemented a very high-resolution topography into the preprocessor to improve the model performance. Some biases particularly in the dynamics suggests uncertainties associated with other factors e.g. interaction between local circulation due to slope winds and synoptic-scale flow, or the representation of highly complex topography of the Himalaya, as correctly pointed out by the reviewer. This study is therefore first step and would be followed up with testing of individual physics schemes as new field measurements become available. These aspects and outlook have been discussed in the revised version (Page: 20-21 Lines 373-406; Page: 25-27 Lines: 503-505, 509-514). It must be however stressed that a model evaluation does qualify for GMD(D) publications as mentioned in the journal's policy.

**Comment 1**: An issue is that when they compare model grid values with that of AWS, they might use two temperature at different height. Please compare the AWS elevation and the grid elevation where AWS located. Use the elevation difference to adjust the model temperature. The same problem also happens to wind speed. There are many evaluation papers for the mountain numerical simulation. The authors should review these papers, try to improve the wind speed performance.

Response 1: Thanks for the valuable suggestion. The difference between actual elevation of the observation site with model grid is 588 m in d01, 480 m in d02, and 270 m in d03 respectively. As the objective here is to describe the improvements in the model output over finer resolutions, we have analyzed model output without adjustments first. Nevertheless, following reviewer's suggestion and following other mountain modelling papers (e.g. Mues et al., 2018), meteorological data adjusted for elevation has also been analyzed in the revised manuscript (Page: 20, Lines: 371-390 and revised Table 1).

**Comment 2**: Figure 3, add their difference between d01 and Radiosonde and give some introduction on the difference. Line 261, it's better to add a figure which shows the correlation coefficient r, mean bias etc. result for all the height, not only say model captures variations at 500 hPa better than 50 hPa. It is also possible to compare the r and mean bias profiles with the three spatial resolution simulation.

Response 2: As suggested, difference between d01 and radiosonde are analyzed and discussed (revised Figure 3; and Page: 13-14, Lines: 269-282). Correlations at different altitudes are presented in form of Taylors diagram (Figure 7a). Following reviewer's suggestion, results summarizing the mean bias, root mean square error, correlation of profiles at different resolution have also been included in the revised version (new Figure 5, and Page: 15-16, Lines: 307-331).

**Comment 3:** Figure 4, many things are not clear in the figure, which year? It also repeat with figure 3. Again, the difference is more interesting to us.

Response 3: Following reviewer's suggestions, Figure 4 (as well as Figure 3) have been modified for clarity. Year (2011) has been mentioned on the revised figure. As suggested, differences are presented in both the figures in revised version.

**Comment 4:** Figure 6 the figure legend is not clear at all. Replot the figure with a colored marker. **Response 4:** As suggested, Figure 6 (Figure 7 in the revised version) has been replotted with proper color marker and legend.

**Comment 5:** Figure 7 where is (a) and (b) letters? what does "0-6-12-.30" mean in the first wind-rose diagram? then why 0-2, 2-4, 4-6.....legend appears on the right of the fourth diagram?

Response 5: Figure 7 (Figure 8 in the revised version) has been revised to address reviewer's comment. Frequency of the occurrence and detailed legends are included now. Previously,

the "0-6-12-...30" was percentage frequency and legend "0-2, 2-4, 4-6....." at the right of the figure was showing the wind speed (ms-1).

**Comment 6**: Figure 8, the simulation does not show the diurnal variation in wind speed at all. What's the explanation for it? This is really interesting for mountain numerical simulation.

Response 6: We agree that the model does not capture the diurnal variation in the wind speed, as also seen over another complex terrain – such as the Tibetan Plateau (Zhou et al., 2019). The daytime reduction in the wind speed was observed by Solanki et al. (2019) over the same mountain peak attributed to the evolution of mountain circulation due to the heating of the slopes and its interaction with the synoptic scale flow, resulting in increased intensity of turbulences and vertical exchange of the momentum fluxes within the surface layer of atmosphere which inhibit the synoptic scale flow up to a certain extent during the daytime. Such competing effect between the thermal and mechanical driven processes could remain unresolved in the model even at higher grid resolution. In addition, mountain winds show sensitivity to boundary layer schemes Yver et al. (2013). Here, we analysed first the impacts of improved representation of the topographical features which would be followed up with testing of different physics schemes in the future. The interpretations with references as well as the limitations and outlook is added in the revised version of the manuscript (Page: 20-21; Lines: 371-406).

**Reply to comments of Reviewer#2**

Authors would like to thank the anonymous reviewer for thorough evaluation of our manuscript and constructive comments. Point-by-point responses to the reviewer's comments are given below in bold fonts and corresponding changes in the manuscript have been highlighted in red color.

**General comment:** This study uses WRF v3.8.1 to explore the effects of spatial resolution on local meteorology. It is very interesting that they found the finer spatial resolution can reduce the biases in simulated meteorology and improve representation of CH through domain feedback into regional-scale simulations. However, in this study, there are too many descriptions of the simulation, but no enough physical explanation to the simulation. It's difficult to make sense that why it occurred. In my view, this manuscript still needs major revision before it can be accepted.

Response: Thanks for the suggestion. Here, we mainly show that more realistic representation of the highly complex terrain, through finer resolution implemented with 3s terrain data leads to better local meteorology of the central Himalaya. Following reviewer's suggestion more discussions including physical explanations have been presented in the revised version of the manuscript, as described in response to specific comments.

**Comment 1:** Section 2.2: How do you process the different temporal resolution of datasets, using the mean value or instantaneous value?

Response 1: Collocated instantaneous values between model and observations have been compared. This is mentioned in the revised manuscript (Page: 9; Lines: 188-189).

**Comment 2:** Line 182-184: It is available of ERA interim at  $0.125 \times 0.125$ , but it's the interpolation results, which may not represent the true performance of ERA interim, especially over the complex terrain regions. It's better to add the comparison between WRF and ERA interim at  $0.75 \times 0.75$ , even there is much less grids of ERA in D03.

Response 2: As suggested, ERA interim at 0.75 x 0.750 has been used for comparison in the revised manuscript (Figure 2; Pages:10-12; Lines: 204; 245-246).

**Comment 3:** Please update the figure captions: i.e., units of all the variables in Figure 3; caption of Figure 6 is not clear (Fig 6a is the comparison between WRF simulation at D01 and the observation?); Figure 8 is only focused D01, etc. you should make them clear in figure caption.

Response 3: Thanks for pointing this out. We have revised the figure and provided clear caption with details of units. Radiosonde and model d01 is also marked clearly (please see Figures 3, 7a, and 9 in the revised version).

**Comment 4:** Line 259-262: Why did it happen? The different vertical distribution and the lower correlation at lower altitudes mostly come from the influence of land-air interaction. Please discussing the possible factors of your results.

Response 4: We agree that the interactions of the underlying surface with lower troposphere profoundly affects the dynamics and local circulations. In mountainous terrains, most important interactions include slope winds and the synoptic scale flow (Solanki et al., 2019). Orographic drag has been suggested to be additional source of the lower correlation (Zhou et al., 2018, 2019). This is discussed in the revised version of the manuscript (Page: 16 Lines: 312-322; Page: 20-21 Lines: 382-406).

**Comment 5:** How do you process the different spatial representation of different simulation and observation? For D01, one grid can indicate the mean situation of 15\*15km area; meanwhile, for D02, it only indicates that in 5\*5km area, etc. please show details of your methods to compare the grid simulation and the in-situ observation.

Response 5: The nearest grid point to the observational site is used for comparison (Page: 9; Lines: 188-189) (e.g. Mues et al., 2018; Singh et al., 2016).

**Comment 6:** Line 286-287: It's very interesting that WRF shows a warm bias south side of Himalaya. Many previous studies pointed that there is obvious cold bias over Tibet (including Himalaya), i.e., Zhou et al. (2017) and Gao et al. (2015). Did you check your location of observation site and WRF grids? The warm bias in your WRF simulation is due to the lower terrain height of the grids than the Observed, please check if they are located over valley and the observed located over ridge

Response 6: Thanks for valuable suggestion. The observation site is a mountain ridge. We performed further analysis of model output by accounting for the altitude difference through linear interpolation of the meteorological parameters to the actual altitude of site in the revised version (Figure 6d-f, Table 1). Altitude adjusted data of model shows cold bias in

agreement with Gao et al., (2015). This is discussed in the revised version of the manuscript (Page: 20-21; Lines: 371-399).

**Comment 7:** Figure 5: as the WRF resolution increasing, the diurnal cycle simulation of T and RH are better, but it didn't work for wind speed. please check the location of the WRF grids and observed station, if both them located valley or ridge? Besides, Zhou et al. (2019) stressed the importance of turbulent orographic form drag (TOFD) on the diurnal cycle simulation of wind speed. It's better to give more explains of inconsistent diurnal cycle of wind between simulation and observations.

Response 7: The processes such as local circulation, slope wind interaction with the synopticscale flow are the key factors governing the diurnal winds over mountain ridge, as shown in Solanki et al., (2016, 2019). We agree with reviewer's view that turbulent orographic form drag (TOFD) could modify the diurnal evolution of wind over such terrains (Zhou et al (2019). These all aspects with relevant references have been included in the revised version of the manuscript (Page: 20-21; Lines:382-406).

**Comment 8:** RH is also dependent on Temperature. What's the performance of the WRF in simulation Specific Humidity (Q)? Please compare the Q between WRF simulation and observation.

Response 8: Comparison of specific humidity between model and observations has been investigated (new Figure S1 in the Supplement). The specific humidity (Q2) shows the explicit dependent on the horizontal grid resolution the bias decreases with increasing the grid resolution. The Q2 shows better correlation (0.67 for d01, 0.72 for d02, and 0.77 for d03) than RH (0.43 for d01, 0.45 for d02 and 0.52 for d03). This is discussed in the revised version of the manuscript (Page: 18; Lines: 350-354).

**Comment 9:** Section 3.4: What are the effects of feedback on the wind direction? In WRF-WF experiments, there are obvious difference among the simulated wind direction at three resolutions. Is there any improvement in the WRF-F experiments?

Response 9: The slight improvement in the wind direction is observed in WRF-F, such improvements are explained in the corresponding section 3.4 (Page 24, Line 459-462), where the effect of the feedback is discussed and changes can be seen in the wind rose plot as shown in Figure S2 and S3. Nevertheless, smaller changes were seen in correlations for WS10 (by 0.05) and T2 (by -0.02) (Figure S2). Variations in wind speed and direction shows an improvement in dominant flow direction e.g. easterly, westerly and north-westerly (Figure S3).

**Comment 10:** Section 3.5: You should check the orographic variation in WRF model output, when you input different geographic data. multi-scale orographic variations are key factors of Wind and moisture simulation over complex terrain, i.e., south side of Himalaya (Wang et al., 2020).

Response 10: The orographic variations in WRF model output have been checked for different geographical input data (Figure 11, Figure S6). As suggested, the spatial

distribution of relative humidity is included in the revised version of the manuscript (new Figure S6; Page:27-28; Lines:525-529). The impact of the orographic variation with different resolution topographic data in RH (Figure S6) shows the differences are in range of -1 to 1% in d02 and -3% to 3% in d03. Such variations are due to inclusion of the SRTM3s high resolution topographic data allowing the model to capture more variation, such orographic features are seen to impact the distribution of moisture in line with suggested study (Wang et al., 2020).

**1 Effects of spatial resolution on WRF v3.8.1 simulated meteorology over the central**

2 Himalaya

- 3 Jaydeep Singh1, Narendra Singh1\*, Narendra Ojha2, Amit Sharma3, a, Andrea Pozzer4, 5, Nadimpally
- 4 Kiran Kumar6, Kunjukrishnapillai Rajeev6, Sachin S. Gunthe3, V. Rao Kotamarthi7
- 5 1Aryabhatta Research Institute of Observational Sciences, Nainital, India
- 6 2Physical Research Laboratory, Ahmedabad, India

[revised manuscript text omitted]

---

## Referee Report (RR1)

This study is helpful to understand the WRF performance of finer resolution over the region of complex terrain. However, this manuscript contains a lot of detailed errors and spelling/unit/font inconsistency. It still needs further improvement before publication. More detailed comments are as follows:

1) Section 1 (introduction): There has been many studies using higher resolution in Himalayas, i.e., 10km (Zhou et al., 2018; 2019), 3km (Wang et al., 2020), and many other studies of high resolution modelling around Himalaya regions.
2) Please introduce the altitude adjustment methods of P, T2, RH2 and WS10. List their equations and references.
3) The modelling in finer resolution has main advantage at resolving orography, not only altitude impacts. But the altitude differences between the simulations of course (or fine) resolution and the observation indeed show impacts on the evaluation metric values, especially for T2. You can compare air temperature in more layers (i.e., 10 layers) near surface in d01, d02, and d03 simulations.

---

## Author Response (AR2)

**Response to Reviewer # 1**

Authors would like to thank the anonymous reviewer for thorough evaluation of our manuscript and constructive comments. Point-by-point responses to the reviewer's comments are given below in bold fonts, and corresponding changes in the manuscript have been highlighted in red.

**General comment:** This study is helpful to understand the WRF performance of finer resolution over the region of complex terrain. However, this manuscript contains a lot of detailed errors and spelling/unit/font inconsistency. It still needs further improvement before publication. More detailed comments are as follows:

**Response: The points raised by the reviewer have been meticulously addressed and incoprporated wherever required in the manuscript.**

**Comment 1:** Section 1 (introduction): There has been many studies using higher resolution in Himalayas, i.e., 10km (Zhou et al., 2018; 2019), 3km (Wang et al., 2020), and many other studies of high resolution modelling around Himalaya regions.

**Response: Thanks for suggesting the important studies, we have referred all in the intrduction. (Page: 5; Line: 106).**

**Comment 2:** Please introduce the altitude adjustment methods of P, T2, RH2 and WS10. List their equations and references.

**Response: The altitude adjustment is basically a linear interpolation of the vertical profiles of meteorological parameters, to the actual altitude (pressure) of the station, instead of taking output at model surface level. The equation is included in the supplementary material**

**(Equation 4). This standard method has been used in previous studies. e.g. Mues et al., 2018**

**It is now mentioned in revised manuscript. (Page: 21; Line: 389-392)**

**Comment 3:** The modelling in finer resolution has main advantage at resolving orography, not only altitude impacts. But the altitude differences between the simulations of course (or fine) resolution and the observation indeed show impacts on the evaluation metric values, especially for T2. You can compare air temperature in more layers (i.e., 10 layers) near surface in d01, d02, and d03 simulations.

**Response: We agree with the reviewer. As suggested, the comparison of temperature in three domains for more different layers (3th, 6th, and 10th) is shown below.**

[Figure]

**As expected, the diurnal amplitudes are smaller at higher model layers. Additionally, the differences among different domains (d01, d02, and d03) decrease at higher layers. This figure has been added in the supplement (Figure S4), and this information is added in the revised version of the manuscript (Page: 21; Line: 395-398).**

**Response to Reviewer # 2**

Authors would like to thank the anonymous reviewer for thorough evaluation of our manuscript and constructive comments. Point-by-point responses to the reviewer's comments are given below in bold fonts, and corresponding changes in the manuscript have been highlighted in red color.

**General comment:** This manuscript analyzes a high-resolution WRF simulation of the 2011 monsoon season for which there are radiosonde observations from a field campaign (GVAX) for comparison. Three domains are set up at 15, 5, and 1 km, centered at the GVAX site in the central Himalaya. And a fourth domain at 333-m grid spacing is simulated for one month within the longer simulation. With increasing resolution, the WRF domains match the observations increasingly well. But it is only in the 333-m domain that the distribution of winds is at all well represented, due to the inadequate representation of the topography in the coarser domains. This study is novel in (1) the various diagnostics it calculates for its multiple domains; (2) the comparison of model results to an intensive field campaign; (3) downscaling to a 333-m domain and the associated 3s resolution SRTM topography data that resolves individual peaks and valleys. I don't see any major issues with the manuscript, but I do have a large number of minor suggestions that I would suggest the authors take into account before submitting a revision. In addition, the English language needs to be improved, although the grammatical errors do not detract from the readability of the manuscript. I have detailed all the grammatical errors up to the end of section 2, but thereafter they are too numerous to list. I suggest enlisting a native speaker.

**Response: Thanks for pointing out novelty and providing the valuable inputs. The manuscript is revised thoroughly, and the language have also been substantially improved.**

**Specific comments**

- Including the version of WRF that you used in the manuscript title is perhaps a bit specific.

**This is done in line with GMD policy which suggests including the version number in the title of the manuscript. (https://www.geoscientific-model-development.net/about/ manuscript_types.html#item2).**

-L19: anthropogenic pressure → stress from anthropogenic forcing

**The suggested change has been made (Page: 1; Line: 18-19).**

- L35: "highlighting significance of well-resolved terrain effects" → ", highlighting the importance of well-resolved terrain"

**The suggested change has been made (Page: 2; Line: 35-36).**

- L51: Spell out GVAX at first use.

**Abbreviation is expanded at first use (Page: 1; Line: 23).**

- L74: Delete "Of late". This is redundant in the sentence.

**The suggested change has been made (Page: 4; Line:74)**

- L88–91: This sentence reads as if High Mountain Asia and the western U.S. are the only places where WRF has been used over complex terrain. I suggest rephrasing to: "WRF has been used for model experiments over complex terrain around the world, e.g., the Himalayan region..."

**We agree with reviewer's observation and the statement has been rephrased (Page: 4; Line:88-89).**

- L102–104: The last sentence of this paragraph is quite obvious. I would suggest deleting it.

**Suggestion has been incorporated.**

- L105–109: This paragraph is very misleading. You claim that most of the modeling studies over the Himalaya have been at 30–45 km grid spacing, except the Mues et al. study that you cite. But there are in fact many studies that have employed WRF at high resolution over the Himalaya, e.g., Potter et al. 2020 (and references within), Norris et al. 2020 (and references within), and Cannon et al. 2017 (and references within). I think this is an opportunity to better emphasize the novelty of your study. I think your study is novel in (1) the various diagnostics it calculates for its multiple

domains; (2) the comparison of model results to an intensive field campaign; (3) downscaling to a 333-m domain and the associated 3s resolution SRTM topography data that resolves individual peaks and valleys. However, this paragraph suggests that the novelty of your study is that it is one of the first to use WRF at 5 km to 1 km grid spacing over the Himalaya, which it is not.

**The discussion has been revised in line with reviewer's comments. Previous studies pointed out by the reviewer have been included in the revised version (Page: 5; Line: 105-117). Additionally, the novelty of our study has also been emphasized (Page: 5; Line: 105-117), as suggested.**

- L107–109: Related to previous comment: "...it remains unclear how the finer resolution could better resolve the complex terrain...". This sentence suggests an ignorance of all of the previous modeling studies over the Himalaya at convective-permitting grid spacing. Your study is novel in the 333-m domain that you employ (although there may be others that have gone down to this grid spacing over the Himalaya). You should either remove this sentence, or stress here that you are running WRF at sub-kilometer resolution.

**We agree with the reviewer's comment and this sentence is removed.**

- L110–111: Related to previous comments: I suggest deleting: "With this opportunity of model evaluation and improvements in simulating meteorological and dynamical variability over the CH, here".

**Suggestion has been incorporated.**

- In Fig. 1, the axes in the top panel should be labeled "Longitude" and "Latitude" to match the other panels.

**The suggested change has been made (Page: 8; Figure 1)**

- L171–172: This sentence is redundant and can be deleted.

**The sentence is removed from the text (Page: 8).**

- L174: You should justify why these dates were chosen, i.e., to match the observation period.

**The simulation period is chosen considering the availability of continuous surface observations from 11 June 2011 and to allow sufficient spin-up time of 10 days for the model to achieve its equilibrium state (Angevine et al., 2014; Seck et al., 2015; Jerez et al., 2020). This is mentioned in the revised version (Page: 8-9, Line: 172-174).**

- Section 2.2: You should refer to Fig. 1 for the location of the station and state its elevation.

**Suggestion is incorporated (Page: 9; Line:186-187)**

- L204–206: This sentence should be in the methodology section.

**Suggestion is incorporated (Page: 6; Line: 129-132)**

- L206: "available at 0.75 x 0.75" is repetition of the previous sentence.

**The redundancy is removed.**

- You define the IGP acronym in multiple places. Just need to define it once.

**Suggestion incorporated in the revised text. (Page: 7; Line: 154-155)**

- L225: Delete "and it could be"

 **Suggestion has been incorporated. (Page 11, Line: 24)**

 - L226: profiles → maps

**Suggestion has been incorporated. (Page: 11; Line: 225)**

- L226: "significantly distinct meteorology" is very vague. What do you mean exactly?

**We agree with the reviewer. In order to avoid the ambiguity, the statement has been rewritten. (Page: 11; Line: 225-227)**

- In Fig. 2, I would suggest two or three appropriately chosen elevation contours so that the meteorological features can be related to the topography. Same in Fig. 10.

**Three elevation contours at 500m (dashed), 1500m (thin black), and 2000m (thick black) lines are included in this Figure 2 (Page: 12; Line: 233-234) and Figure 10 (Page: 26; Line:**

**480-481). The description is added in the revised manuscript (Page: 10, 12; Line: 207-208, 251-253).**

- L236: distinct → increasingly distinct

**Suggestion has been incorporated. (Page: 12; Line: 236).**

- L258: The phrase "overestimated as compared to ERA-Interim" is highly misleading because WRF is probably much more accurate than ERA-Interim. I would suggest rephrasing the sentence to reflect this.

**We agree. The sentence is rephrased as "higher variability than that in the ERA-interim (1.2–2.3ms$^{-1}$) due to finer resolution of the WRF" (Page: 13; Line: 250-261).**

- L258–259: What are the two datasets: WRF and ERA-Interim? This last sentence of the paragraph appears to contradict the preceding text in which you highlight the differences.

**This sentence is rephrased as – "Overall, the impact of the topography resolved at higher resolution in WRF shows the contrasting differences in surface pressure, temperature, relative humidity and wind speed compared to coarse resolution ERA-interim dataset" (Page: 13; Line: 261-263).**

- At the beginning of section 3.2, it would help to reiterate the location and elevation of the radiosonde observations (shown in Fig. 1).

**As suggested, the location and elevation of the observational site are included (Page: 13; Line: 265-266).**

- L267: Delete "(c)" (repetition).

**Suggestion has been incorporated. (Page: 14; Line: 271).**

- Fig. 3: I would suggest giving the wind speed profiles the same x axis values as other panels, i.e., missing data for the observations up to the beginning of September, so that features in wind can be easily related to features in other variables. Same in Fig. 4.

**We appreciate reviewer's suggestion, however, the vertical profiles of wind from radiosonde are missing for two months which is showing a large part of the Fig 3 and 4 as blank, so we intend to retain the comparison as it is. For this purpose, Figure S2 in supplementary can be referred and discussed in the text (Page: 14; 289-291). The exercise has been carried out to understand the association among meteorological parameters on a common scale as given in the modified Figure4 below, but it doesn't show a significant feature to be considered. Therefore, we chose not to include in the manuscript.**

[Figure]

**Modified Figure 4**

- Fig. 3: You should clarify in the axis titles of panels g—i that these panels show WRF minus the observations ("difference" is ambiguous).

**As suggested, "WRF-observations" has been marked on the color scale (revised Figure 3) to remove ambiguity.**

- L272: The temperature inversion can only be seen on certain days and requires zooming in to see clearly. Maybe regions of negative dT/dp could be marked in the figure?

**The zoomed version of the Figure 3a, d which shows the contour plots of the vertical profiles of temperature.**

[Figure]

**The region where dT/dt is negative is shown by two different contour lines (black: -0.05 $^0$C/hPa and magenta: -0.2 $^0$C/hPa). The temperature inversion is clearly depicted in this figure. The gradient increases with altitude which is the characteristic of this region. This figure is also placed in the supplementary text (Figure S1) and description is added in the revised manuscript (Page: 14; Line: 276-277).**

- L273: 3a → 3d

**Suggestion has been incorporated. (Page: 14; Line: 277).**

- L274: this variability → these features

**Suggestion has been incorporated. (Page: 14; Line: 279).**

- L279: "wetter (or more humid)" → "more humid"

**Suggestion has been incorporated. (Page: 14; Line: 283).**

- L279: dry bias → low-humidity bias

**Suggestion has been incorporated. (Page: 14; Line: 284).**

- L288: "normalized standard deviation (SD), normalized to the standard deviation of observation" → "standard deviation, normalized by that of the observations"

**The sentence is rephrased to "standard deviation, normalized by that of the observations". (Page: 15; Line: 295)**

- L290: "which turns out to be less than 1" → ", less than 1"

**Suggestion has been incorporated. (Page: 15; Line: 297).**

- L293: 50 hPa → 75 hPa (unless the figure legend is wrong?). Related to this point, relative humidity in the stratosphere is not really relevant. It's probably more appropriate to quote the value at 250 hPa in Fig. 7, i.e., an indication of deep convection.

**Thanks for pointing out this typo. The actual value was 75 hPa, instead of 50hPa. The values indicated in the brackets are the correlation coefficients (r), not the relative humidity. Text is modified. (Page: 15; Line: 298, 300).**

- L299: Delete "below 500 hPa (Figure 4)" (repetition).

**Suggestion has been incorporated.**

- L303: How can you tell that the model qualitatively captures the vertical profiles if only the differences between the observations and model are plotted?

**We agree with the reviewer. The text has been slightly modified. However, the comparison of model simulated meteorological parameters with the radiosonde is also analyzed and the statistical metrics (correlation, standard deviations etc.) are compared in Figure 7a. (Page: 15; Line: 298-300, 307-310).**

- L328–330: It is worth mentioning that the higher-resolution models do worse in the upper troposphere for RH? Can you say why this is?

**We agree with the reviewer's observation and mentioned in the revised version(Page: 17; Line: 336-337). The possible reasons for poor performance of the models in upper troposphere could be a) stronger convection in the model at higher resolution, b) the Current WRF-set up is using a non-local boundary layer schemes in this case and c) the moisture parameter itself is highly variable with altitude and becomes almost negligible few kms after crossing the zero degree isotherm d) the performance of radiosonde/instrument sensors in very low temperatures may also degrade the model-sensor relationship. However, these case studies can be interestingly taken up in future as an independent exercises.**

- L348: Figure 6b → Figure 6

**Suggestion has been incorporated. (Page: 19; Line: 357).**

- L353: Delete "lower by" (repetition).

**Suggestion has been incorporated. (Page: 19; Line: 362).**

- L354: What is Figure 7bs? Do you just mean Figure 7b?

**Typo error has been corrected in the revised version (Page: 19; Line: 363).**

- L358: RH → RH2

**Suggestion has been incorporated. (Page: 19; Line: 368).**

- In Table 1, Clarify what the +- sign refers to. And clarify "min/max" - is this the min and max during the full observation period?

**These ambiguities have been resolved in revised caption of Table 1. Min / Max is the minimum and maximum during the full observation period. (Page: 19; Line: 374)**

- Fig. 6 caption: Explain error bars for observations. Why no error bars for model results? Same in Fig. 9.

**Error bars represent the standard deviation and were shown only for observations for clarity of figure (and standard deviation values for simulations are given in the Table 1). Nevertheless, in revised version, error bars are also shown for a model result (d01) (revised**

**Figure 6). Similarly, standard deviation values for Figure 9 are shown in Table 2 and are shown for d01 in revised version. (Page: 21; Line: 383-384)**

- L380: few → a few

**Suggestion incorporated.**

- L382: What does 0.35 refer to?

**0.35 is the correlation coefficient value for 2m temperature (T2) between the model and observation. It is also mentioned in the manuscript. (Page: 21; Line: 393)**

- L396–411: Where are the numbers quoted in this paragraph shown?

**In this paragraph, we have compared our finding (shown in Table 1 and Table S1) with the values estimated in various studies for which the references are mentioned. (Page: 22; Line: 410:421). We have clarified this in the revised version.**

- L397: -0.7 → -0.7 g/kg

**Suggestion has been incorporated. (Page: 22, Line: 411-412).**

- L425–443: In this paragraph, you begin stating percentages, e.g., "The dominance of the wind flow over the observational site is easterly (30%)". Then, you switch to the frequency, e.g., "The frequency of southerly (539) and south-westerly (SW, 481)..." You should pick one, either percentage or frequency, and stick to it to make the text consistent.

**The wind flow dominance over the observational site is easterly (30%) and south-easterly (26%) while 26% occurrences of wind are from the west and north-west. The percentage of southerly (21%) and south-westerly (SW, 19%) are relatively higher in d01 as compared to the observations (S: 4% and SW: 5% respectively) ....". (Page: 24; Line 444-447)**

- L433: Add "and westerly" after "north easterly". (The overestimate of westerlies in d03 in Fig.8 should be noted.)

**The suggested changes are incorporated in the revised manuscript. (Page: 24; Line: 449)**

- L435: You quote 27% for south-easterlies in the observations. But earlier you quote 26% (L430). Which is it? The sentence on L434–435 could actually be deleted because it is repetition of a few sentences prior.

**It is actually 26% and to avoid the repetition L434-435 has been deleted.**

- L438: "over predicted by 10%" is misleading. What you mean is "over predicted by 10 percentage points".

**The sentence is rephrased as per the suggestion. (Page: 24; Line: 452)**

- L439: Insert "in the observations" after "very weak" if I understand correctly?

**The text has been modified and presented accordingly. (Page: 24; Line: 453-454).**

- L439–441: Where is it shown that the wind direction changed during transitions from high to low wind conditions?

**This statement has been modified accordingly. (Page:24; Line: 454-456).**

- L450: Why do you quote Figs. 2 and 6 specifically. It seems that this is shown in all figures?

**We agree, it is not required and removed.**

- L452: daughter → nested. Same on L477.

**Suggestion has been incorporated. (Page: 25, 27; Line: 467, 495).**

- L458: and the effect→ showing the effect

**"and the effect" is replaced by "showing the effect" (Page: 25; Line: 473).**

- Fig. 10: Right hand panel title should be "2m Relative Humidity (%)"

**Thank you for pointing it out. The title of the right hand panel is changed from "Relative Humidity (%)" to "2m Relative Humidity (%)". (Page: 26; Figure 10)**

- L478–479: How can you say that the effect of feedback is strikingly observed? Can you be more specific?

**The spatial variabilities of 2m temperature and relative humidity significantly changes by inclusion of the feedback over mountainous region which is discussed in text (Page: 27; Line:495-496). However, it is rephrased in revised version of manuscript (Page: 27; Line: 494-495).**

- L479: Why "the trend of diurnal variation of the relative humidity"? Don't you just mean "relative humidity"?

**The sentence is rephrased (Page: 26; Line:480).**

- L489: Delete "(30s or ~0.95km)" (repetition).

**Suggestion has been incorporated. (Page: 27; Line: 507).**

- In section 3.5 when you introduce d04, it would be very helpful for the location of d04 to be shown within d03 in Fig. 1 (bottom right panel).

**The d04 domain is included in Figure 1, and the caption of the figure is updated accordingly (Page: 8; Figure 1). This is also included in the text (Page: 27; Line:512)**

- L502–503: After this sentence it would help to quote the elevation of the station location in d03 versus d04.

**The elevation of the location is quoted as per the suggestion. (Page: 28; Line: 519-520)**

- L505: Add "(Figure 11c)" after "d04".

**Suggestion has been incorporated. (Page: 28; Line: 525)**

- L506: "which could be smoothed out if the simulation was carried out with GMTED / or at 1 km with SRTM3s" → "which are smoothed out with GMTED (Figure 11a) or with SRTM3s interpolated to 1 km (Figure 11b)"

**The sentence is rephrased, as suggested. (Page: 28; Line: 526-527)**

- L508–513: Where are all the values that are quoted in these lines shown? Is there a supplementary figure that should be referenced?

**The MB in wind speed are calculated from the model and corresponding observations (Figure 12). A supplementary Figure S5 and Table S2 is added where these numerical values are mentioned. (Page: 28; Line: 528-531)**

- L519: The sentence: "The southerly...model resolution" is confusing. Do you mean" The frequency of southerlies increases with model resolution, better matching the observations"?

**The sentence is rephrased. The result obtained is that the frequency of southerly component matches better with that of the observations in case of SRTM3s than GMTED (Page: 28-29; Line: 537-538).**

- L522: "while occurrence of south-easterly winds is underestimated" → "with the exception of south-easterly winds"

**Suggestion has been incorporated. (Page: 29; Line:540-541)**

- Fig. 12. Why are the frequency distributions not shown here, like in Fig. 8? It seems Fig. S5 could be incorporated into Fig. 12?

**The frequency distribution plot Figure S5 is incorporated in Figure 12 (Page: 29).**

- L550: pressure → surface pressure

**Suggestion has been incorporated. (Page: 30; Line-571).**

- L557: wind directions → frequency distribution of wind direction

**Suggestion has been incorporated. (Page: 30; 578-79).**

**Grammatical errors (up to end of section 2)**

- L20: Himalaya → the Himalaya (and in other places)

**Suggestion has been incorporated. (Line: 20, 47, 58, 74 etc.).**

- L28: the northern → northern

**Suggestion has been incorporated. (Line: 28).**

- L32: surface → the surface

**Suggestion has been incorporated. (Line: 32).**

- L32: the d01 → d01 (and in other places)

**Suggestion has been incorporated. (Line: 32).**

- L33: "and are closer" → ", closer"

**Suggestion has been incorporated. (Line: 33).**

- L34: coarser simulation → the output from the coarser domains

**Suggestion has been incorporated. (Line: 34).**

- L34: station → the station

**Suggested change has been made (Line: 35).**

- L37: nested domain → the nested domain

**Suggested change has been made (Line: 38).**

- L37: demonstrated → demonstrates

**Suggested change has been made (Line: 38).**

- L42: was → is

**Suggested change has been made (Line: 42).**

- L42: "a frequent southeastward wind component remained underestimated" → "the frequency of south-easterlies remains underestimated"

**Suggested change has been made (Line: 42-43).**

- L47: Himalayan region → The Himalayan region

**Suggested change has been made (Line: 47).**

- L51: called as → known as

**Suggested change has been made (Line: 51).**

- L55: "hydrological cycle especially monsoon system" → "hydrological cycle, especially the monsoon system"

**The sentence is rephrased as suggested by the reviewer (Line: 55).**

- L67: terrains → terrain (and in other places)

**Suggested changes have been done (Line: 67, 72, 84, 226 etc).**

- L76: Further → Consequently

**Suggested change has been made (Line: 76).**

- L76: have → has

**Suggested change has been made (Line: 76).**

- L77: in past → in the past

**Suggested change has been made (Line: 77).**

- L79: their causes range from mesoscale processes to larger synoptic scale events" → "the associated weather systems range from mesoscale to synoptic-scale phenomena"

**The sentence is replaced as suggested by the reviewer (Line: 79)**

- L80: observational network → an observational network

**Suggested change has been made (Line: 80).**

- L82: would → can

**Suggested change has been made (Line: 83).**

- L84: Himalaya → the Himalaya

**Suggested change has been made (Line: 84).**

- L88: "WRF model" → "WRF" or "The WRF model" (and in other places)

**Suggested change has been made (Line: 88, 98, 104 etc.).**

- L90: "intermountain west of the United States" → "the multiple mountain ranges in the western United States"

**The sentence is rephrased as suggested by the reviewer (Line: 91)**

- L100: correctly → accurately

**Suggested change has been made (Line: 101).**

- L101: "Kathmandu valley of Himalaya" → "the Kathmandu valley"

**Suggested change has been made (Line: 102).**

- L102: "attributed to complex topography → ", which they attributed to insufficient resolution of the complex topography,"

**The sentence is rephrased as suggested by the reviewer (Line: 103-104)**

- L102: "3 km x 3 km" → "3 km"

**Suggested change has been done (Line: 104).**

- L118: larger Indian region → the larger Indian region

**Suggested change has been made (Line: 113).**

- L122: Subsequent → The subsequent

**Suggested change has been made (Line: 118).**

- L131: prediction → Prediction

**Suggested change has been made (Line: 127).**

- L132: Eulerian → an Eulerian

**Suggested change has been made (Line: 128).**

- L147: "an explicit micrcophysics scheme is needed to resolve cloud and precipitation processes" → "cloud and precipitation processes are resolved by the microphysics (MP) scheme"

**The sentence is rephrased as suggested by the reviewer (Line: 143-144)**

- L158: Outer → The outer

**Suggested change has been made (Line: 155).**

- L158: north → the northern

**Suggested change has been made (Line: 154).**

- L159: "Himalayan mountains with vegetation and snow cover" → "and the Himalayan mountains" (why do you need to mention the vegetation and snow cover?)

**"with the vegetation and snow cover" is removed from the text and the sentence is rephrased as suggested by the reviewer (Line: 155)**

- L160: "Model atmosphere has 51 vertical levels" → "The model has 51 atmospheric vertical levels"

**The sentence is rephrased as suggested by the reviewer (Line: 157)**

- L164: "innermost domain 03" → "the innermost domain, d03,"

**Suggested change has been made (Line: 160).**

- L165: mainly to reveal → to resolve

**Suggested change has been made (Line: 161).**